# Adult stem cells and niche cells segregate gradually from common precursors that build the adult *Drosophila* ovary during pupal development

**Amy Reilein\*†, Helen V Kogan†, Rachel Misner, Karen Sophia Park, Daniel Kalderon\***

Department of Biological Sciences, Columbia University, New York, United States

**Abstract** Production of proliferative follicle cells (FCs) and quiescent escort cells (ECs) by follicle stem cells (FSCs) in adult *Drosophila* ovaries is regulated by niche signals from anterior (cap cells, ECs) and posterior (polar FCs) sources. Here we show that ECs, FSCs, and FCs develop from common pupal precursors, with different fates acquired by progressive separation of cells along the AP axis and a graded decline in anterior cell proliferation. ECs, FSCs, and most FCs derive from intermingled cell (IC) precursors interspersed with germline cells. Precursors also accumulate posterior to ICs before engulfing a naked germline cyst projected out of the germarium to form the first egg chamber and posterior polar FC signaling center. Thus, stem and niche cells develop in appropriate numbers and spatial organization through regulated proliferative expansion together with progressive establishment of spatial signaling cues that guide adult cell behavior, rather than through rigid early specification events.

**\*For correspondence:**
areilein@gmail.com (AR);
ddk1@columbia.edu (DK)

†These authors contributed
equally to this work

**Competing interest:** The authors declare that no competing interests exist.

## Introduction

Adult stem cells generally act as a community, coordinating the activities of multiple individual cells to replenish tissues throughout life (*Clevers and Watt, 2018*; *Post and Clevers, 2019*). Their behavior is invariably regulated by external signals from other cell types, known as niche cells (*Biteau et al., 2011*; *Losick et al., 2011*; *Voog and Jones, 2010*). It is therefore essential to establish adult stem cells in suitable numbers and niche environments during development to ensure appropriately regulated community activity in adults. Understanding the underlying developmental processes would facilitate identification of genetic developmental defects that disrupt adult tissue repair or confer cancer susceptibility, and creation of expandable organoids maintained by stem cells for therapeutic use, drug studies, or mechanistic investigations (*Akbari et al., 2019*; *Merenda et al., 2020*; *Petersen et al., 2018*; *Pourquie et al., 2018*; *Yousef Yengej et al., 2020*; *Drost and Clevers, 2017*).

The development of terminally differentiated adult cell types commonly results from a series of successive, definitive, and largely irreversible decisions that are stabilized by potentially long-lasting shifts in gene expression and chromatin modification (*Adam and Harwell, 2020*; *Davidson et al., 2002*; *Rothman and Jarriault, 2019*; *Sagner and Briscoe, 2019*; *Salinas et al., 2020*; *Zechner et al., 2020*). However, the development of adult stem cells has not been studied extensively and presents a number of special considerations. First, stem cells typically continue to proliferate in adulthood and often exhibit malleable relationships with their products, with the potential for marked changes in inter-conversion rates in either direction (*Donati and Watt, 2015*; *Tetteh et al., 2015*; *Wabik and Jones, 2015*). Does the general paradigm of definitive cell-type specification at a fixed time in development still apply, despite this marked flexibility of the final

product? Second, do adult tissues and the stem cells that maintain the tissue derive from separate or common precursors? Third, is there coordination between the development of adult stem cells and their niche cells, potentially serving to arrange appropriate proportions and juxtaposition of the two cell types?

*Drosophila* ovarian follicle stem cells (FSCs) present a particularly interesting and experimentally favorable paradigm to investigate because the organization, behavior, and regulation of these adult stem cells have been studied thoroughly, revealing many similarities to the important mammalian paradigm of gut stem cells (*Gehart and Clevers, 2019*; *Hayashi et al., 2020*; *Reilein et al., 2017*). FSCs directly produce two different types of cell in adult ovaries, follicle cells (FCs) and escort cells (ECs) (*Reilein et al., 2017*). Both of these cell types also act as niche cells, producing key JAK-STAT and Wnt signals, respectively (*Reilein et al., 2017*; *Melamed and Kalderon, 2020*; *Song and Xie, 2003*; *Vied et al., 2012*; *Waghmare and Page-McCaw, 2018*). Similarly, Paneth cells in the mammalian small intestine are both stem cell products and niche cells (*Beumer and Clevers, 2020*; *Hsu and Fuchs, 2012*; *Fuchs and Blau, 2020*). FSCs also provide a window into sustaining a tissue supported by two different types of stem cell. The mechanisms of coordination among FSCs, germline stem cells (GSCs), and their products in the adult are not well understood but must rely on an organization that is established during development (*Hayashi et al., 2020*; *Waghmare and Page-McCaw, 2018*; *Kahney et al., 2019*; *Gilboa, 2015*).

GSCs and FSCs are housed in the germarium, which is the most anterior region of each of the 30–40 ovarioles comprising the adult ovary (*Figure 1A*; *Hayashi et al., 2020*). Somatic terminal filament (TF) and cap cells at the anterior of the germarium are stable sources of niche signals for GSCs and FSCs, including bone morphogenetic proteins (BMPs), Hedgehog, and Wnt proteins, with GSCs directly contacting cap cells (*Hayashi et al., 2020*; *Kahney et al., 2019*). Cystoblast daughters of GSCs divide four times with incomplete cytokinesis to yield a progression of 2-, 4-, 8-, and finally, 16-cell germline cysts. Their maturation is accompanied by posterior movement and depends on interactions with neighboring quiescent, somatic ECs (*Kirilly et al., 2011*). ECs also act as niche cells for adjacent or nearby GSCs and FSCs (*Hayashi et al., 2020*; *Waghmare and Page-McCaw, 2018*; *Sahai-Hernandez and Nystul, 2013*; *Wang and Page-McCaw, 2018*).

A rounded stage 2a 16-cell cyst loses EC contacts as it forms a lens-shaped stage 2b cyst spanning the width of the germarium and subsequently recruits FCs (*Figure 1A*). The earliest FCs can be recognized by expression of high levels of the surface adhesion protein, Fasciclin 3 (Fas3) (*Reilein et al., 2017*; *Zhang and Kalderon, 2001*). The stage 2b cyst rounds to a stage 3 cyst as surrounding FCs proliferate to form an epithelium and segregate specialized polar and stalk cells, which allow budding of the FC-enveloped cyst as an egg chamber from the posterior of the germarium (*Figure 1A*; *Duhart et al., 2017*). Polar cells produce the JAK-STAT ligand, Unpaired, which patterns the development of FCs along the anterior-posterior (AP) axis of each developing egg chamber (*Duhart et al., 2017*; *McGregor et al., 2002*; *Xi et al., 2003*) and creates a gradient of signaling activity across the FSC domain, which is critical for FSC divisions and for FSC conversion to FCs (*Melamed and Kalderon, 2020*; *Vied et al., 2012*). Egg chambers bud roughly every 12 hr in each ovariole of well-fed flies, progressively enlarge and mature towards egg formation and deposition, as they move posteriorly.

There are about 16 FSCs that reside in three rings or layers around the inner germarial surface between ECs and FCs on the AP axis (*Figure 1A*; *Hayashi et al., 2020*; *Reilein et al., 2017*). The most posterior FSCs directly become FCs, while anterior FSCs directly become ECs, but FSCs also exchange AP locations, allowing a single FSC lineage to include both ECs and FCs (*Reilein et al., 2017*). Individual FSCs are frequently lost and frequently amplify, with no evident temporal or functional link between division and differentiation (*Reilein et al., 2018b*), so that the representation and output (of ECs and FCs) of any one FSC lineage is highly variable (*Reilein et al., 2017*). All of these behaviors are largely regulated by extracellular signals that are graded along the AP axis, with especially prominent roles for JAK-STAT ligand derived from polar cells and Wnt ligands derived from both cap cells and ECs (*Reilein et al., 2017*; *Melamed and Kalderon, 2020*; *Song and Xie, 2003*; *Vied and Kalderon, 2009*; *Wang et al., 2021*). The number of FSCs, together with the spatially regulated signals that guide division rate, is matched to steady-state production of 5–6 FCs every 12 hr (*Reilein et al., 2018b*), while the number and location of ECs is important not only to deliver some niche signals directly but may also be important to keep FSCs at a suitable distance from the cap cell source of Wnt and Hh signals.

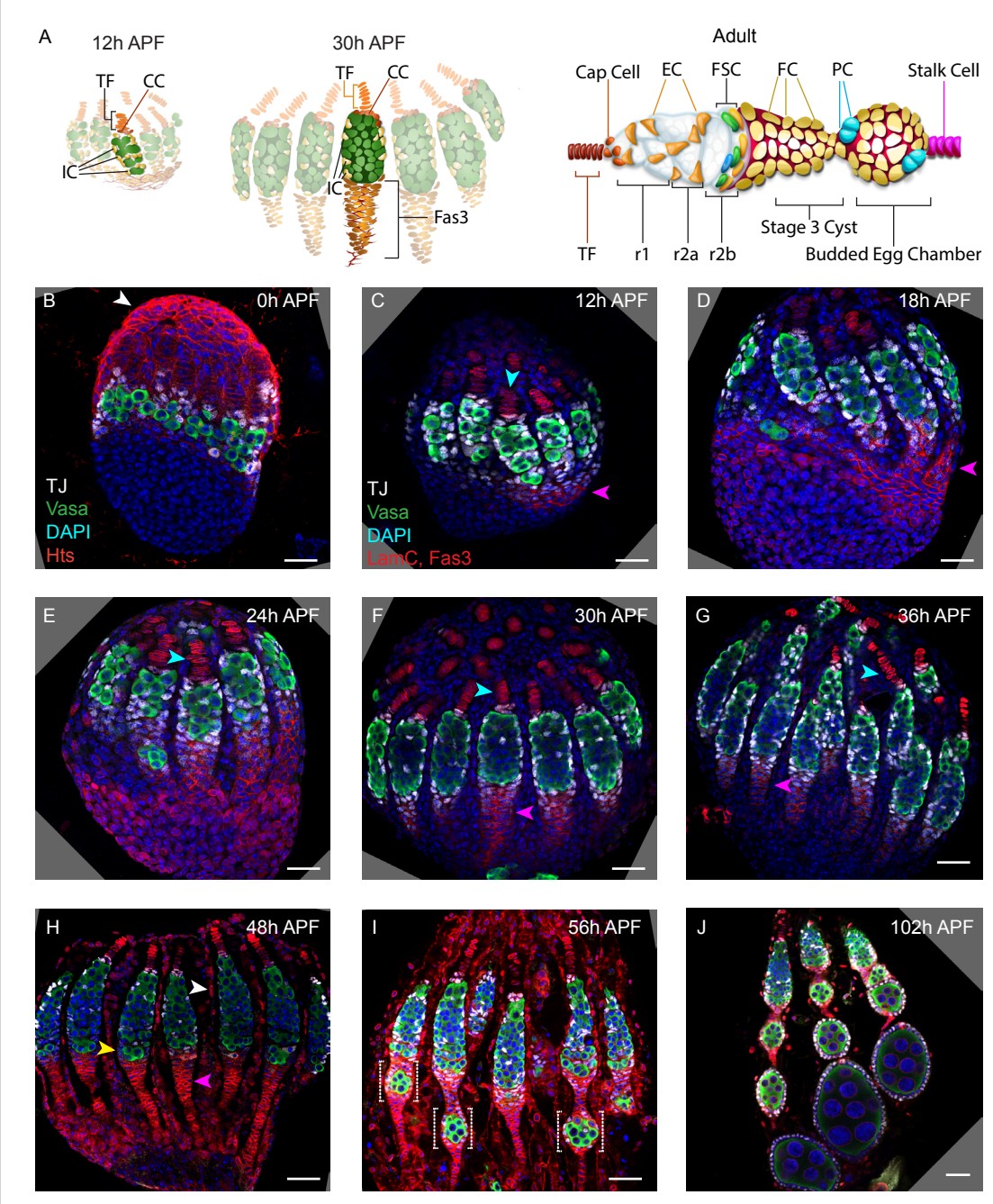

**Figure 1.** Development of ovarioles during early pupal stages. (**A–J**) A time course of pupal ovary development. (**A**) Cartoons highlight individual germaria from 12 hr after puparium formation (APF), 30 hr APF and adults, showing terminal filament (TF), cap cells (CC), and intermingled cells (IC). In the adult germarium, region 1 (**r1**) and region 2a (r2a) escort cells (ECs; about 40 in total) surround developing germline cysts (white) and three rings of follicle stem cells (FSCs; about 16 in total) line the germarial circumference. Follicle cells (FCs) all initially express the surface protein Fas3 (red outlines), giving rise to an epithelium of an egg chamber, stalk cells that separate egg chambers, and polar cells (PC), the source of the Upd ligand that activates JAK/STAT signaling to stimulate FSC proliferation and conversion to FCs. (**B–J**) Vasa (green) marks germline cells, Traffic Jam (TJ, white) marks somatic ICs and DAPI (blue) marks all nuclei. (**B**) Hts (red) marks apical cells. (**C–J**) Lamin-C (red) marks TF cells anterior to ICs (cyan arrowheads in **C–G**), while Fas3 (also red) marks cells posterior to ICs that form a basal stalk (pink arrowheads in **C–G**). All scale bars, 20 µm. (**B**) At 0 hr APF, ovaries are not yet organized into distinct germaria. Individual germline cells (green) are intermingled with somatic IC precursor cells (white). The medial side of the ovary (right) has more developed TF. 'Apical cells,' the precursors of epithelial sheath cells, are located at the apical (top) side of the ovary (white arrowhead). (**C**) At 12 hr APF, ovaries are organized into distinct germaria, separated by a layer of migrating apical cells. Lamin-C stains the nuclear envelope of TF cells (cyan arrowhead). Fas3 staining is present on the medial side of the ovariole (pink arrowhead). TJ staining of somatic cells extends slightly beyond the germline. (**D**) At 18 hr APF, more cells express Fas3 (red; pink arrowhead), and (**E**) by 24 hr APF, Fas3 is spread more evenly around the circumference

*Figure 1 continued on next page*

*Figure 1 continued*

of the ovary, and stains a hub into which the posterior ends of nascent basal stalks converge. (**F, G**) By 30–36 hr APF, individual germaria are farther apart, separated by epithelial sheath cells that express Lamin-C weakly, and basal stalks (pink arrowheads in **F–H**) begin to narrow. (**H**) By 48 hr APF, Fas3 begins to be expressed dimly in posterior ICs (yellow arrowhead) and Lamin-C expression (red) is clear in epithelial sheath cells (white arrowhead). (**I**) At 56 hr APF, germaria begin to bud the first egg chamber and (**J**) by 102 hr APF, 3–4 egg chambers have budded. See also *Figure 1—figure supplements 1–2*.

The online version of this article includes the following video and figure supplement(s) for figure 1:

**Figure supplement 1.** Germline development during early pupal stages.

**Figure supplement 2.** Pattern of division of germline and somatic cells in pupal ovaries.

**Figure 1—video 1.** Full z-section stacks of a 0h APF ovary.
https://elifesciences.org/articles/69749/figures#fig1video1

**Figure 1—video 2.** Full z-section stacks of a 12h APF ovary.
https://elifesciences.org/articles/69749/figures#fig1video2

**Figure 1—video 3.** Full z-section stacks of a 18h APF ovary.
https://elifesciences.org/articles/69749/figures#fig1video3

**Figure 1—video 4.** Full z-section stacks of a 24h APF ovary.
https://elifesciences.org/articles/69749/figures#fig1video4

**Figure 1—video 5.** Full z-section stacks of a 30h APF ovary.
https://elifesciences.org/articles/69749/figures#fig1video5

**Figure 1—video 6.** Full z-section stacks of a 36h APF ovary.
https://elifesciences.org/articles/69749/figures#fig1video6

**Figure 1—video 7.** Full z-section stacks of a 48h APF ovary.
https://elifesciences.org/articles/69749/figures#fig1video7

**Figure 1—video 8.** Full z-section stacks of a 56h APF ovary.
https://elifesciences.org/articles/69749/figures#fig1video8

Previous studies have outlined how germline and somatic ovary precursors are first selected during embryogenesis, then migrate and coalesce to form a nascent gonad (*Jemc, 2011*; *Murray et al., 2010*), followed by specification of anterior TF cells and separation into individual developing ovarioles during larval development (*Gilboa, 2015*; *Godt and Laski, 1995*). Understanding of pupal development (*Godt and Laski, 1995*; *Cohen et al., 2002*; *Irizarry and Stathopoulos, 2015*; *King et al., 1968*; *Valer et al., 2018*) was limited, until recently, by technical difficulties associated with dissecting and imaging pupal ovaries (*Park et al., 2018*). Here we use morphological examination of developing pupal ovaries through fixed and live images, together with cell lineage studies, to examine how adult FSCs, ECs, and FCs are produced and organized from precursors present at the end of larval stages.

Marked lineages were clustered along the AP axis, with larger clones towards the posterior, suggesting limited AP dispersion and an AP gradient of proliferation. FSC lineages almost always included marked FCs and mostly included ECs, especially if initiated early, indicating that specific stem cell precursors are not determined early; instead, FSCs likely simply acquire their characteristics according to AP location at the end of pupal development. Combined lineage and morphological studies showed that ECs and FSCs, as well as some FCs, derive from intermingled cell (IC) precursors that are interspersed with germline cells throughout pupal development, while an expanding population of cells posterior to ICs become the FCs of the first-formed egg chamber. Live imaging showed that the first pupal egg chamber is formed by migration of a germline cyst out of the developing germarium into an accumulation of future FCs, contrasting with the emergence of an FC-encased germline from the germarium that characterizes adult egg chamber budding. Thus, in this paradigm, stem cells are not specified at a fixed time in development. They arise from precursors that also build the niche and the adult tissue they support. The use of delayed specification and shared precursors, together with an evolving spatial gradient of proliferation, may be an effective general developmental strategy to juxtapose and produce appropriate numbers of stem and niche cells in the adult tissue.

## Results

### Outline of ovary development

We dissected and imaged developing ovaries during pupation to provide a systematic descriptive overview, informed by prior studies (*Figure 1*). Germaria within each ovary develop from medial to lateral and the overall duration of pupation can vary significantly, leading to some differences in the reported timing of critical stages (*Cohen et al., 2002*; *Lai et al., 2017*; *Li et al., 2019*). We assign '0 hr after puparium formation (APF)' as the time when larvae cease moving (pupariation). Animals mostly eclosed 5 days later (120 hr APF) and the first egg chamber had budded from about half of all germaria slightly before 60 hr APF (*Figure 1I*).

Early ovary organization progresses from anterior (generally termed apical at early stages) to posterior ('basal') (*Figure 1*; *Gilboa, 2015*; *Godt and Laski, 1995*). By the end of the third larval instar, each ovary has 15–20 anterior stacks of non-dividing TF cells (*Figure 1A and B*), which express Lamin-C but not Traffic Jam (TJ) (cyan arrowheads in *Figure 1C and E–G*; *Godt and Laski, 1995*; *Panchal et al., 2017*; *Sahut-Barnola et al., 1996*). The most posterior 'basal cells,' which do not express TJ (*Figure 1B*), arise from swarm cells that migrated from anterior locations prior to the late larval stage (*Banisch et al., 2021*). 'Apical cells' (white arrowheads, *Figure 1B and H*), initially anterior to the TF stacks, migrate towards the posterior (basally), first separating each TF stack and then entire developing ovarioles over the next 48 hr as they form an epithelial sheath lined by a basement membrane (*Cohen et al., 2002*). Fas3 expression initiates in a subset of cells on the medial side of the ovary (pink arrowheads, *Figure 1C and D*; *Lai et al., 2017*) and is later found in cells of epithelial appearance that intercalate progressively from posterior to anterior, to form a 'basal stalk' (pink arrowheads, *Figure 1F–H*) at the posterior end of each developing ovariole (*Figure 1A*; *Godt and Laski, 1995*; *Vlachos et al., 2015*).

Developing germaria consist of germline cells (green in *Figure 1B–J*) and somatic cells expressing TJ (white in *Figure 1B–J*) between the TFs and basal cells. About six of these somatic cells surround the most posterior TF cell, the transition cell, and become non-dividing cap cells, which express Lamin-C and TJ, shortly after the larval/pupal transition in response to Notch signaling and an ecdysone pulse (*Panchal et al., 2017*; *Gancz and Gilboa, 2013*; *Gancz et al., 2011*; *Song et al., 2007*; *Yatsenko and Shcherbata, 2018*; *Figure 1A*). Primordial germline cells (PGCs) that contact cap cells largely remain in position to become adult GSCs, supported by local cap cell products, including BMPs and Perlecan (*Diaz-Torres et al., 2021*; *Asaoka and Lin, 2004*; *Zhu and Xie, 2003*). The remaining PGCs develop similarly to adult cystoblasts to produce germarial cysts and four egg chambers during pupation.

A population of somatic cells posterior to cap cells in the developing germarium are interspersed amongst germline cells and were consequently named intermingled cells (ICs) (*Figure 1A*). ICs express TJ (white in *Figure 1B–J*) but not Lamin-C or Fas3 (both red in *Figure 1C–J*) at pupariation (*Gilboa, 2015*; *Lai et al., 2017*; *Panchal et al., 2017*). ICs have been suggested to be the source of adult ECs (*Gilboa, 2015*), but neither this relationship nor the source of FSCs and FCs in newly eclosed adults has been clearly defined experimentally.

### Spatially organized progression of germline differentiation

We are principally interested in understanding the developmental origins of somatic ECs, FSCs, and FCs, but this process must be coordinated with germline development to produce egg chambers. Posterior PGCs start to express the early differentiation marker, Bam-GFP, just prior to pupariation, coincident with a strong ecdysone pulse (*Gancz et al., 2011*; *Zhu and Xie, 2003*). Branched fusomes are seen shortly afterwards, indicating divisions with incomplete cytokinesis to form cysts with two or more cells (*Li et al., 2019*; *Gancz et al., 2011*), but the subsequent progression of germline differentiation during pupation has not been described. We therefore examined germline development by using Hts antibody (red in *Figure 1—figure supplement 1*) to reveal a rounded spectrosome, characteristic of single germline cells, or a branched fusome, which connects Vasa-marked germline cells (green in *Figure 1—figure supplement 1*) within a cyst (*Deng and Lin, 1997*), and by incubating samples with EdU to assess DNA replication as a proxy for cell division (*Figure 1—figure supplement 2*).

We observed  2 cell cysts at 0 hr APF, cysts of up to 8 cells by 12 hr APF, and 16 cell cysts at 18 hr APF (*Figure 1—figure supplement 1A–C*). Progressively larger cysts were arranged, with few exceptions, from anterior to posterior from 18 hr APF onward; this was particularly clear in samples from 24

to 48 hr APF (*Figure 1—figure supplement 1D–F*). A pair of posterior cysts were often side by side in germaria at 24 hr APF but only a single cyst occupied the most posterior position in most cases by 36 hr APF, and in all germaria by 48 hr APF (*Figure 1—figure supplement 1D–F*). Consistent with these observations of cyst progression and location, EdU labeling was often seen in the most posterior cyst(s) up to 24 hr but rarely at 36 hr, while at 48 hr even the second most posterior cyst in the germarium rarely incorporated EdU (*Figure 1—figure supplement 2A–D*). Our observations reveal a progression of germline differentiation from single cells to 16 cell cysts that is spatially organized along the AP axis during the first 48 hr APF; at 48 hr APF, the organization of germline cysts resembles that in the adult up to a stage 2b cyst (*Figure 1A*).

## FSCs and ECs derive from a shared precursor

We conducted a series of lineage analyses to learn about the specification of ECs, FSCs, and FCs from precursors during pupal development (*Figure 2A*). In these studies, we collected newly eclosed adults that had been heat-shocked during development. The timing of clone initiation was deduced from the time elapsed between administration of a heat-shock to induce *hs-flp*-mediated genetic recombination and adult eclosion. A '–5d sample,' generated by collecting adults centered around 120 hr after heat-shock reflects cell marking at around 0 hr APF (*Figure 2A*) because the time from pupariation to eclosion was generally a few hours less than 120 hr, but there is also a short delay between heat-shock and genetic marking of a cell (*Golic and Lindquist, 1989*).

Ideally, a single precursor per developing ovariole would be labeled at a variety of times during pupation, followed by a comprehensive analysis of each resultant lineage in large numbers of ovarioles from newly eclosed adults. In practice, it is difficult to label only one cell in a developing ovariole when there are many precursors present. We used a very mild heat-shock protocol to try to accomplish this objective. We principally used MARCM clonal analysis (*Lee and Luo, 2001*) because positive cell marking with GFP greatly facilitates scoring all cells in a lineage. However, maturation of a strong GFP MARCM signal in adult and developing ovarioles requires almost 4d. We therefore dissected ovarioles from adults 2d after eclosion ('+2d') in our first experiments, having induced clones daily (in separate samples) between 6d and 2d before eclosion ('–6d' to '–2d'; *Figure 2A*). Subsequent experiments focused on studying single-cell lineages by using multicolor labeling to reduce the frequency of labeling a precursor in a specific color, and on capturing all labeled FCs produced during pupation by examining lineages in newly eclosed adults (*Figure 2A*).

In the first experiments, involving staged clone induction, each ovariole in 2d-old adults was scored for the number of marked ECs and FSCs (*Figure 1A*) by imaging and archiving complete confocal z-stacks (*Figure 2*). Clones were classified according to inferred cell types present at the time of eclosion (see Materials and methods for extrapolating data from +2d adults to 0d adults). The average number of labeled ECs and FSCs in a marked adult germarium decreased when clones were induced at progressively later stages (from an average of 8.1 at –6d to 2.0 at –2d), consistent with proliferation of most precursors during this period (*Tables 1* and *2*). As expected, TF cells were almost never labeled and cap cell labeling was seen only at the earliest times of induction (present in 3% of clones induced at –5d [0 hr APF]) because these non-dividing cell types have already differentiated shortly before (TFs), or just after (CCs) the larval/pupal transition and the MARCM method only labels dividing cells.

Strikingly, almost all ovarioles with early-induced clones (from –6d to –4d) that included marked FSCs also contained marked ECs ('EC/FSC clones') (*Figure 2B, G and H*; *Table 1*). Ovarioles containing only marked ECs ('EC only') were seen at each time of clone induction (*Figure 2B, E, F, I, J*) while a significant frequency of ovarioles with marked FSCs but no marked ECs ('FSC only') was evident only when induced at 48 hr or 72 hr APF (*Figure 2B and L*). These observations immediately suggested that most adult FSCs derive from a dividing precursor that gives rise to both ECs and FSCs. During the second half of pupation, an increasing proportion of the products of early EC/FSC precursors produce FSCs but no ECs.

## Relative location of precursors along AP axis instructs fate

The marked cells in an adult germarium were generally clustered in the AP dimension (*Figure 2E–L*). To evaluate clustering numerically, we counted marked region 1 (r1) and region 2a (r2a) ECs separately. Region 1 contains germline cysts with fewer than 16 cells and comprises the most anterior

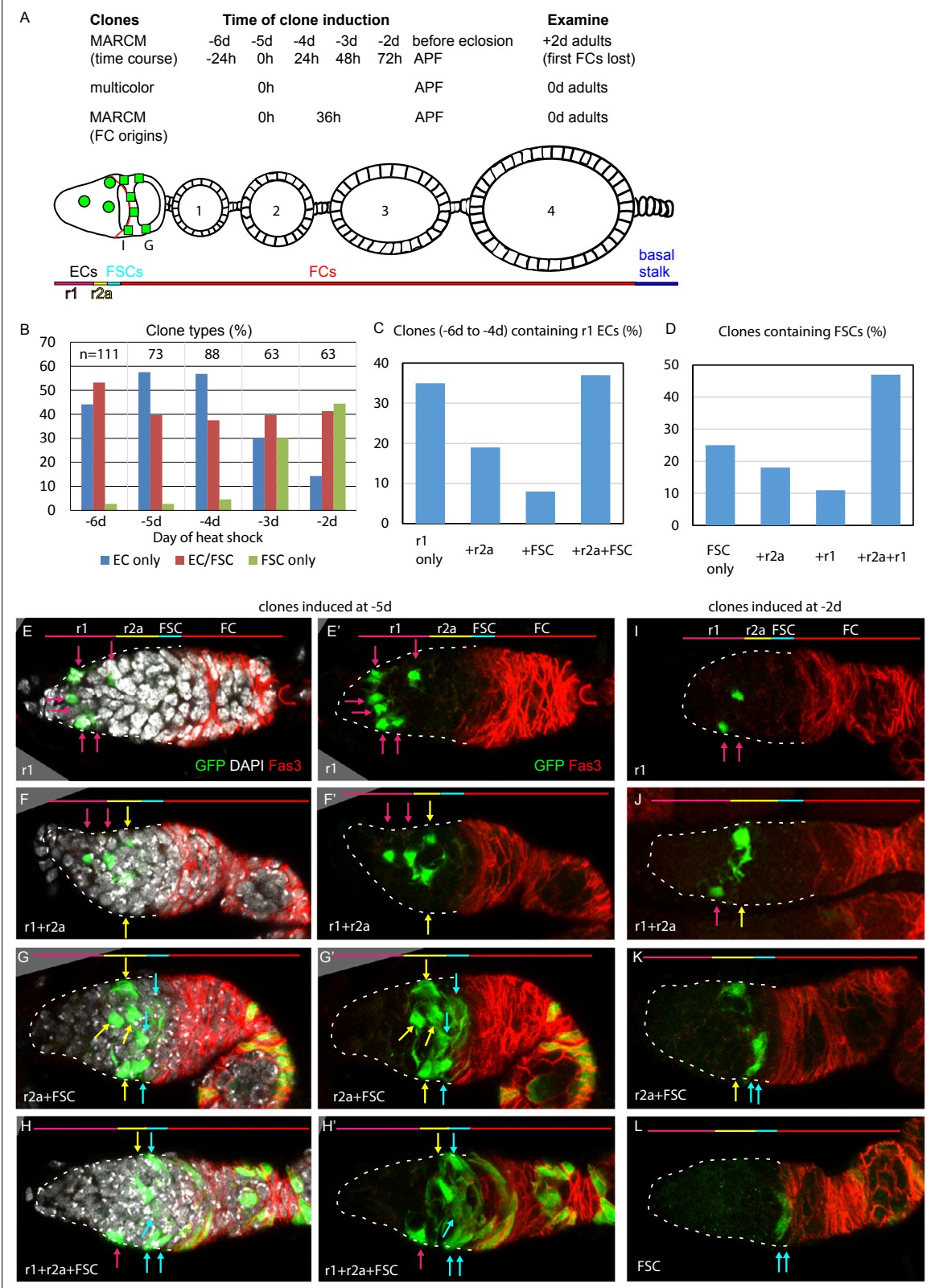

**Figure 2.** Common precursors for follicle stem cells (FSCs) and escort cells (ECs) with derivatives clustered along the AP axis. (**A**) Summary of lineage tracing experiments performed in this study. The ovariole of a newly eclosed adult, which has four egg chambers, is depicted with an example lineage (green cells). In the first set of experiments ('time course'), we dissected ovarioles from adults 2d after eclosion ('+2d'), having induced MARCM clones daily (in separate experiments) between 6d and 2d before eclosion ('–6d' to '–2d'), corresponding to the given times after puparium formation (APF).

*Figure 2 continued on next page*

*Figure 2 continued*

The second approach used multicolor labeling to reduce the frequency of distinctly marked lineages, so that ovarioles rarely contain more than one lineage of the chosen color. The third approach examined ovarioles from newly eclosed adults to capture all labeled follicle cells (FCs) produced during pupation. Labeled cells were scored as r1 (magenta) or r2a (yellow) ECs, FSCs (blue), or FCs (red); color code depicted at the bottom of (**A**) and the top of (**E–L**). Marked FC locations were scored as immediately posterior (I) to the Fas3 border (red) , in the posterior half of the germarium (G) and around each numbered egg chamber (1–4). (**B–L**) GFP-labeled MARCM clones were induced from 6d to 2d before eclosion and flies were dissected 2d after eclosion. In each germarium, labeled cells were scored as region 1 (**r1**) ECs, region 2a (r2a) ECs, or FSCs. (**B**) Most clones initiated 4–6d before eclosion contained only ECs (blue) or FSCs together with ECs (red), indicating a common precursor. Clones containing only ECs declined and clones containing FSCs but no ECs (green) increased in frequency when initiated at later times. (**C, D**) Data were pooled from clones induced from –4 to –6d. (**C**) Clones containing r1 ECs (n = 174) were grouped according to the additional cell types they contained; inclusion of other r1 ECs was the most frequent, and r2a ECs were more commonly included than more distant FSCs. (**D**) Clones containing FSCs (n = 155) contained r2a ECs more frequently than r1 ECs. (**E–H**) Examples of clones induced 5d before eclosion demonstrate the clustering of marked cells (green) along the anterior (left) to posterior (right) axis. The anterior Fas3 (red) border, viewed in each z-section, allows distinction between FCs and FSCs (in the three layers anterior to Fas3); several z-sections were combined to show all labeled cells in one image. DAPI (white) staining reveals all nuclei but is omitted in (**E'–H'**) and (**I–L**) for clarity. GFP-marked clones have (**E**) a cluster of r1 ECs (magenta arrows), (**F**) a cluster of r1 and r2a ECs (yellow arrows), (**G**) a cluster of r2a ECs and FSCs (blue arrows), and (**H**) r1 and 2a ECs with FSCs. (**I–L**) Clones induced 2d before eclosion also showed clustering of progeny, with (**I**) r1 ECs only, (**J**) r1 and r2a ECS, (**K**) r2a ECs and FSCs, and (**L**) FSCs only. Scale bars, 20 μm.

The online version of this article includes the following figure supplement(s) for figure 2:

**Source data 1.** Numerical data for graphs in *Figure 2*.

two-thirds of EC territory along the AP axis (*Figure 2A and E–L*). Of all early-induced clones (–6d to –4d) with two or more marked cells and including at least one r1 EC, 35% contained only r1 ECs, 19% included also only r2a ECs, while only 8% also included only FSCs (the remainder included both r2a ECs and FSCs; *Figure 2C*). Thus, genetic relatives of a marked region 1 EC are often only in region 1; they are also more commonly found in region 2a than in the more distant FSC region, even though the total frequency of clones with marked region 2a ECs (54%) and FSCs (57%) were similar (*Figure 2C*). Conversely, of all clones with two or more marked cells including at least one FSC, 25% contained no ECs, 18% included also only r2a ECs, while only 11% also included only r1 ECs, even though the total frequency of clones with r1 ECs (68%) was higher than for r2a ECs (54%) (*Figure 2D*).

These observations suggest that (i) precursors occupy significantly different AP positions early in pupal development, (ii) their progeny undergo stochastic movement in either direction but still remain relatively close in the AP dimension, and (iii) cell identity (r1 EC, r2a EC or FSC) is set by their AP location at eclosion. By adulthood, this identity includes starkly different behaviors of quiescence (ECs) or proliferation (FSCs). Had precursors been restricted early in pupation to produce only FSCs, only ECs, or only one type of EC (r1 or r2a), marked lineages would be restricted to a single-cell type. In fact, we observed many lineages that spanned boundaries between r1 and r2a ECs, and between ECs and FSCs, especially for clones induced from 0 to 48 hr APF (*Figure 2B–D, F–H, J and K*).

**Table 1.** Frequency of clones originating from different precursors and cellular yields per clone.

| Heat-shock to eclosion (days) | precursors of r1 ECs | | precursors of r1/2a ECs | | | precursors of EC/FSCs | | | | precursors of FSCs | |
|---|---|---|---|---|---|---|---|---|---|---|---|
| | Freq. | Yield | Freq. | Yield | | Freq. | Yield | | | Freq. | Yield |
| | % | r1 | % | r1 | r2a | % | r1 | r2a | FSC | % | FSC |
| 2 | 7.9 | 1.2 | 6.3 | 0.3 | 1.8 | 41 | 0.5 | 1.2 | 1.3 | 44 | 1.2 |
| 3 | 14 | 1.2 | 16 | 0.8 | 2.2 | 40 | 0.9 | 1 | 2.6 | 30 | 1.4 |
| 4 | 34 | 2.1 | 23 | 2 | 1.6 | 38 | 1.6 | 1.5 | 3.7 | 4.6 | 3 |
| 5 | 33 | 2.2 | 25 | 2.8 | 2.3 | 40 | 3.1 | 3.4 | 5.5 | 2.7 | 2.5 |
| 6 | 31 | 3.5 | 14 | 3.3 | 3.6 | 53 | 3.2 | 3.4 | 4.8 | 2.7 | 3 |

EC: escort cell; FSC: follicle stem cell.

**Table 2.** Number of adult cells of each type made from dividing precursors at different developmental times calculated from the total yield of marked cells of each type.

| Heat-shock to eclosion (days) | Total number of marked cells in clones | | | | Cells made per germarium from dividing precursors | | | |
|---|---|---|---|---|---|---|---|---|
| | r1 | r2a | EC | FSC | r1 | r2a | EC | FSC |
| | x | y | x + y | z | 16x/z | 16y/z | 16(x + y)/z | |
| 2 | 20 | 37 | 57 | 69 | 4.6 | 8.6 | 13.2 | 16 |
| 3 | 41 | 46 | 87 | 93 | 7.1 | 7.9 | 15 | 16 |
| 4 | 156 | 80 | 236 | 134 | 18.6 | 9.6 | 28.2 | 16 |
| 5 | 191 | 140 | 331 | 164 | 18.6 | 13.7 | 32.3 | 16 |
| 6 | 356 | 253 | 609 | 295 | 19.3 | 13.7 | 33 | 16 |

EC: escort cell; FSC: follicle stem cell.

The AP clustering of progeny and the derivation of both FSCs and ECs from a common precursor were investigated further by using a multicolor lineage approach, which has the key virtue of reducing the frequency of cell marking with a specific color.

## Low-frequency clones validate a common precursor for ECs and FSCs

The overall clone frequency (percentage of ovarioles with any marked cells in the germarium or early egg chambers) in the above experiments was 60% . It would therefore be expected that about 2/3 of marked ovarioles derive from marking a single precursor if all recombination events were independent and equally likely (see Materials and methods). We then used a multicolor marking strategy, in which GFP-positive clones lacking both lacZ and RFP result from two recombination events and should therefore be especially rare among the six possible color combinations generated (*Figure 3A*; *Reilein et al., 2017*). GFP-only clones, induced at 0 hr APF and scored in newly eclosed (0d) adults, were found at an overall frequency of 26/115 ovarioles, so that an estimated 86% of labeled ovarioles (see Materials and methods) might be expected to have lineages derived from a single GFP-only cell.

The patterns of GFP-only and other multicolor clones were very similar to those observed by MARCM analysis (*Figure 3*), with a single lineage frequently spanning r1 and r2a territory, or EC and FSC domains (*Figure 3B–D*). Among GFP-only lineages that included FSCs, eight also included ECs (*Figure 3B and C*) and only two did not. We can therefore be confident that a single precursor at –5d (0 hr APF) can give rise to both ECs and FSCs and that most FSCs originate from such precursors.

Lineages of different colors in the same germarium were generally offset along the AP axis, presumably reflecting differences in AP location of the initially marked cell. In some cases, the two differently colored lineages likely derive from a single recombination event in a common parent, yielding 'twin-spot' daughters (*Figure 3A, E and F*). These observations graphically confirm the earlier conclusion that marked lineages spread a limited but significant distance along the AP axis, centered on the cell of origin and frequently spanning boundaries between two different adult cell types.

## EC precursor division declines from the anterior at mid-pupation

The MARCM lineage experiments also yield information about when precursors of specific adult cell types stop dividing because only dividing precursors can be labeled by mitotic recombination. For each time of clone induction, we summed the total number of marked cells of each type (FSCs, r1 and r2a ECs) over all clones (*Table 2*). The ratio of marked ECs (r1 or r2a) to FSCs for any one time of clone induction reports the relative number of those cell types produced from dividing precursors. The absolute number of ECs per germarium made from dividing precursors was then calculated by using a value of 16 FSCs per germarium and assuming that FSC precursors, like FSCs themselves, do not cease dividing and can therefore all be labeled by MARCM throughout development (*Table 2*). This calculation is based on scoring all labeled cells over all clones and is therefore equally valid whether the scored cells in an ovariole derive from a single marked cell or from more than one marked cell.

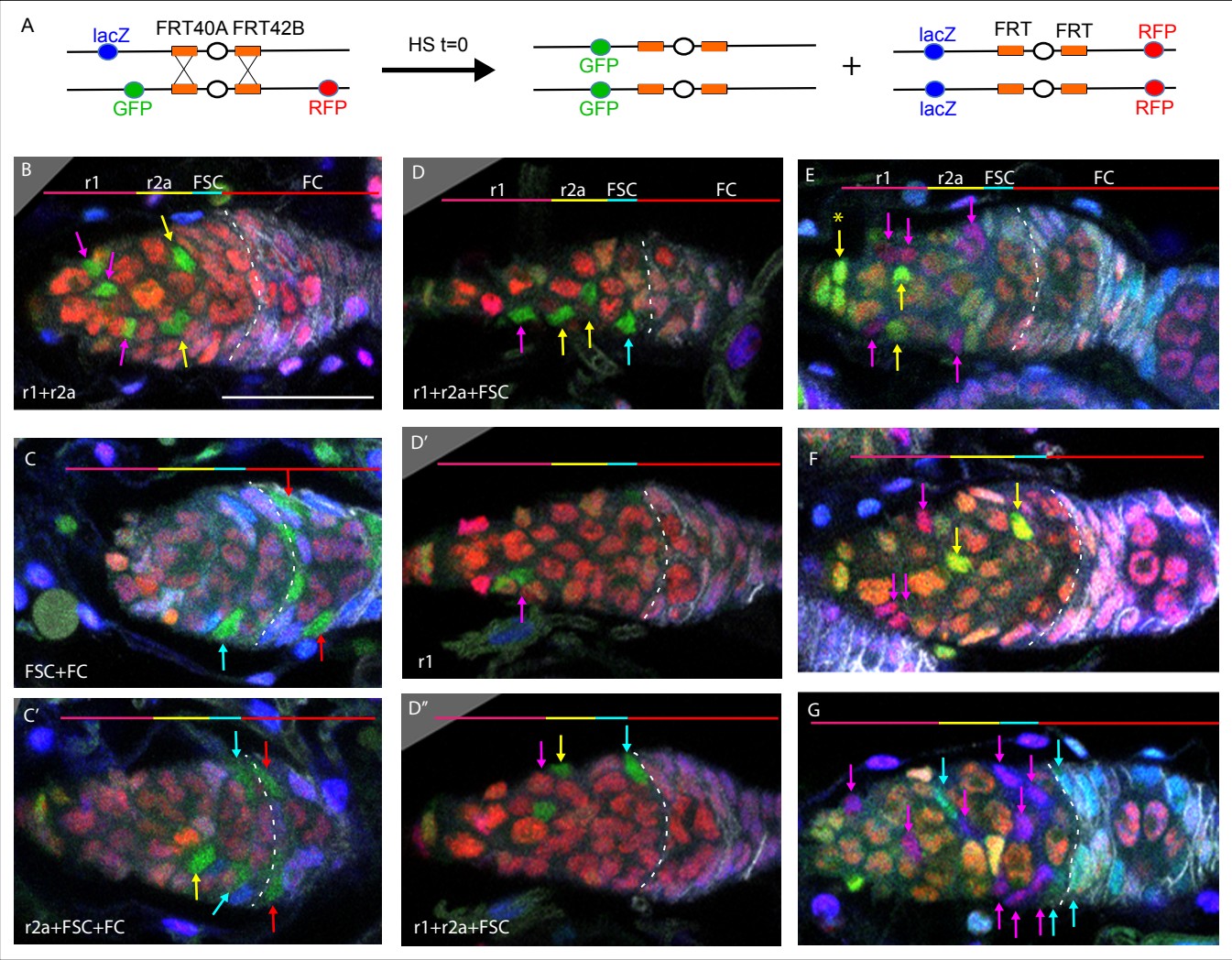

**Figure 3.** Multicolor labeling of precursors shows clustering of progeny along the AP axis. (**A**) Second chromosome genotype of multicolor flies used for multicolor lineage tracing of precursors, showing *tub-lacZ* (lacZ), *ubi-GFP* (GFP), and *His2Av-RFP* (RFP) transgenes and *FRT 40A* and *FRT 42B* recombination targets either side of the centromere (white oval). Heat-shock induction of a *hs-flp* transgene on the X-chromosome can induce recombination independently at either pair of FRTs, making one or both chromosome arms homozygous in daughter cells, thereby eliminating one or more of the marker genes. One possible outcome is shown. One daughter has GFP-only and the other daughter has a complementary genotype (*lacZ* plus RFP); it is generally possible to identify sister 'twin-spot' lineages for other outcomes. (**B–G**) Multicolor lineage tracing was performed by inducing clones 5d before eclosion and dissecting newly eclosed flies. The anterior limit of Fas3 staining is marked by a white dashed line. Scale bar in (**B**) applies to all images, 20 µm. Precursors produce r1 (magenta) and r2a (yellow) escort cells (ECs), follicle stem cells (FSCs) (blue), and follicle cells (FCs) (red) in overlapping zones, indicated by the colored zones and in (**B–D**) corresponding arrow colors. (**B–D**) show examples of GFP-only clones (green), which are generated by two recombination events and are therefore infrequent. (**B**) A clone containing r1 and r2a ECs. (**C, C'**) Different z-planes of a germarium show a clone containing an r2a EC, FSCs, and FCs (only cells present in the z-section shown are listed in white). (**D–D"**) Three z-planes of a germarium showing a clone with r1 and r2a ECs and FSCs. (**E–G**) Examples of two clones of different colors in overlapping or adjacent zones, where arrow color indicates the clone color, not the cell type. (**E**) A GR (GFP plus RFP) clone (yellow arrows) contains r1 ECs and cap cells (extreme left), and a BR (*lacZ* plus RFP) clone (magenta arrows) contains r1 and r2a ECs. (**F**) A BR clone (magenta arrows) contains r1 ECs, and a GR clone (yellow arrows) contains a r2a EC and an FSC. GR and BR clones are potential twin-spot descendants of the two daughters of a parent cell undergoing recombination. (**G**) A BR clone (magenta arrows) contains ECs and FSCs, and a BG (*lacZ* GFP) clone (cyan arrows) contains a r2a EC, FSCs, and FCs.

For –6d and –5d samples, the calculated number of ECs made from dividing precursors was very similar at about 33 (19 r1 and 14 r2a) per germarium (*Table 2*). This approximates the total number of adult ECs, which has been reported to be about 40 on average (*Wang and Page-McCaw, 2018*), and which we counted as 32–51 over 25 germaria from young adults, suggesting that most cells that will become adult ECs are still dividing at the larval/pupal transition. The calculated number of ECs made from dividing precursors dropped slightly in –4d samples (from 33 to 28) and more markedly in –3d

(to 15) and –2d samples (to 13) (*Table 2*). We surmise that precursors of over half of the future adult EC population have ceased dividing by 2–3 days prior to eclosion. Moreover, the average number of dividing precursors per germarium for r1 ECs (4.6) was significantly lower than for r2a ECs (8.6) by –2d (72 hr APF), even though adults contain more r1 than r2a ECs (19 vs. 14, from –6d and –5d data). Thus, cessation of precursor division appears to spread from the anterior towards the posterior as pupal development proceeds.

## Many FCs produced during pupation derive from precursors that also yield FSCs and ECs

In a third cell lineage approach (*Figure 2A*), we induced MARCM clones early enough (0 hr APF and 36 hr APF) that we were able to examine GFP-labeled cells in newly eclosed adults, allowing us to score all FCs produced during pupation, before the earliest FCs are lost in 2d-old adults. FCs typically surround the two most mature germarial cysts and four budded egg chambers in a newly eclosed adult (*Figure 2A*).

We first examined lineages induced at 0 hr APF. Most ovarioles with marked FSCs also had marked ECs (24/27) (*Figure 4A and D*), as observed for MARCM analysis in 2d-old adults and multicolor clone analysis, confirming a common precursor of ECs and FSCs. Almost all clones with marked FSCs also had marked FCs (25/27) (*Figure 4A*), indicating a common precursor for adult FSCs and FCs. A significant proportion of all marked ovarioles (23/98 = 23%) had marked ECs, FSCs, and FCs, indicating that some individual pupal precursors give rise to all three cell types in newly eclosed adults. Some ovarioles ('FC-only') contained marked FCs but no marked FSCs (*Figure 4C and D*). Similar results were observed for multicolor lineage tracing examined in newly eclosed adults (*Figure 3*, *Table 3*). These results suggest that at the start of pupation the most anterior precursors produce ECs, the most posterior precursors produce only FCs and precursors in between produce FSCs, mostly together with both ECs and FCs (summarized in *Figure 5*).

The locations of marked FCs were scored as 'immediate' (region 2b), 'germarial' (region 3), or in egg chambers ('ECh') 1–4 (*Figure 2A* and *Figure 6A*). The percentage of clones with marked immediate FCs was higher for ovarioles that included marked FSCs (56%) than for those without marked FSCs (16%). Conversely, the percentage of ovarioles with marked FCs in at least one of the two terminal (most posterior) egg chambers was higher when marked FSCs were absent (88%) than when they were present (67%). Indeed, occupancy by marked FCs decreased gradually from anterior to posterior for FSC/FC clones and with the opposite polarity for FC-only clones (*Figure 6A*). Thus, derivatives of a given precursor were clustered along the AP axis in FSC/FC territory, as previously noted for EC/FSC territory.

The inference that the fate of precursor derivatives is set largely by the AP position of precursors, with limited AP dispersal of derivatives, appears to be the governing principle all the way from the most anterior r1 EC derivatives to the most posterior FCs of newly eclosed adults (*Figure 5*). Notably, while some clones contained only ECs or only FCs, the AP spread of derivatives generally exceeded the width of the FSC domain, so that ovarioles with marked FSCs always contained ECs or FCs (or both). In other words, there were no FSC-specific precursors at the larval/pupal transition and individual lineages regularly spanned the borders of adult cell identities (*Figure 5*).

## Prevalence of different precursors during pupation

The number of precursors at 0 hr APF that give rise to different adult cell types can be deduced from the observed proportion of lineage categories (EC, EC/FSC ± FC, FSC ± FC, FC) together with the average number of marked FSCs and ECs in each lineage category. However, both measured parameters will be inaccurate if some ovarioles include two or more marked lineages. Analysis of lineages in 0d adults allowed a better estimate of how often this occurred.

In this MARCM experiment, 53% of 207 ovarioles contained no marked cells, suggesting that 72% of marked ovarioles have lineages derived from a single cell if all recombination events are independent (see Materials and methods). However, 16% of marked ovarioles contained marked FCs and marked ECs but no marked FSCs, strongly suggesting that these discontinuous patterns represent two lineages and, hence, that the total proportion of ovarioles with more than one marked lineage is considerably higher than 28%. Thus, there appears to be some clustering of recombination events rather than an even distribution among all ovarioles. The observed frequency of ovarioles with marked

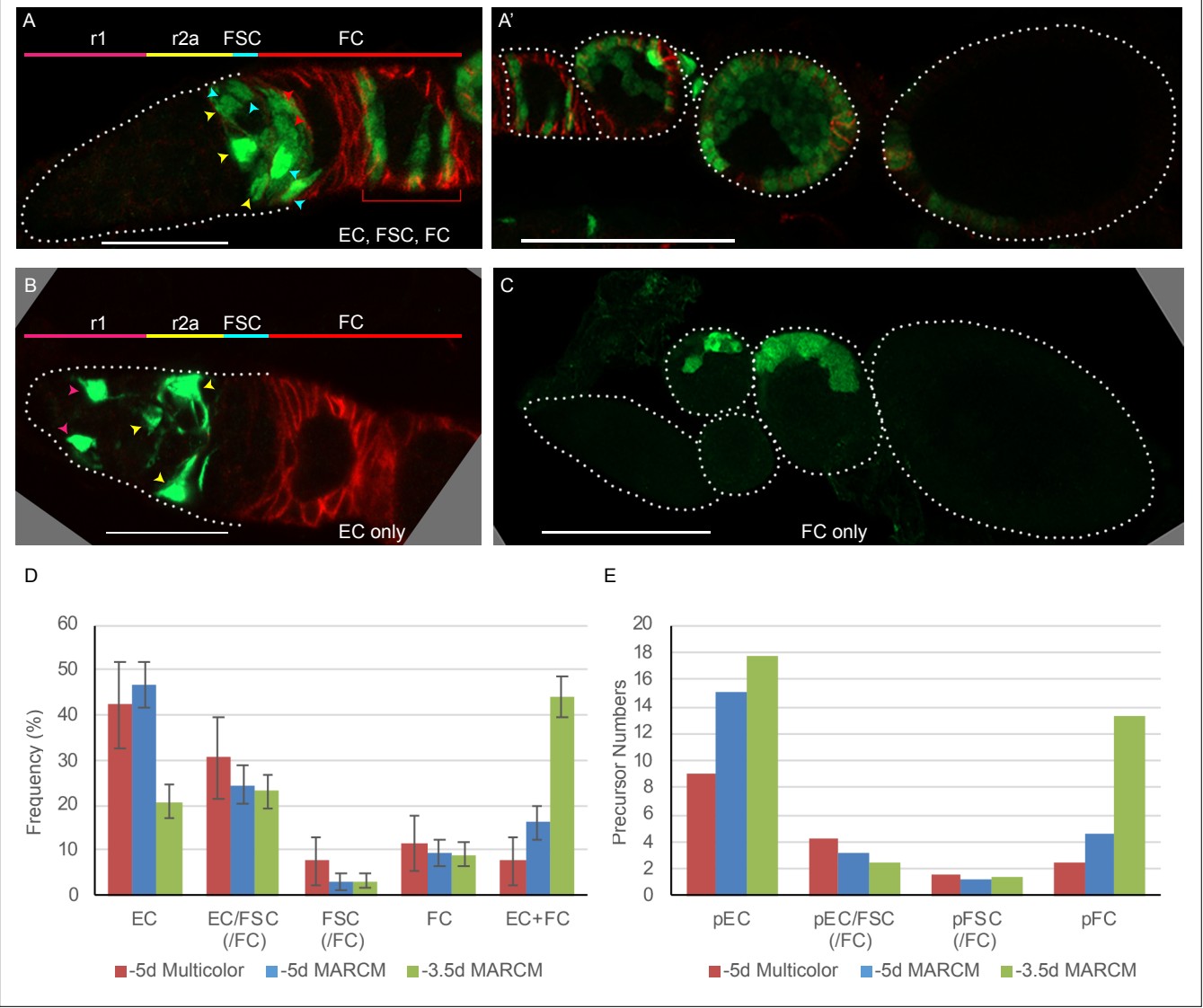

**Figure 4.** Follicle cell origins and deduced precursor numbers for escort cells (ECs), follicle stem cells (FSCs), and follicle cells (FCs). (**A–C**) GFP-labeled MARCM clones (green) were induced 5d before eclosion (0h APF) and ovaries were dissected from newly eclosed adults in order to detect all FCs produced during pupation. Fas3 is stained in red and cell types are indicated with color-coded arrows, matching the labeled AP domains. Scale bar in (**A, B**), 20 μm; (**A', C**), 50 μm. (**A**) Germarium and (**A'**) the rest of the ovariole of the same sample (the first egg chamber is in both images), showing a clone with marked ECs, FSCs, and FCs (marked FCs are in the germarium and egg chambers 1–4). (**B**) A clone containing only ECs (no cells were marked in the rest of the ovariole). (**C**) A clone containing only FCs in egg chambers 2 and 3. (**D**) Distribution of clone types in GFP-only clones of multicolor flies and MARCM clones initiated 5d (0h APF) or 3.5d (36h APF) before eclosion, all examined in newly eclosed adults. Clone types are EC (EC-only), EC/FSC (ECs and FSCs, with or without FCs), FSC (with or without FCs), FC (FC-only), and EC+ FC. Ovarioles with only marked ECs and FCs likely harbor two independent lineages. (**E**) Estimates of numbers of dividing precursor types at the time of clone induction for the same three lineage experiments as in (**D**), after accounting for double clones (see Materials and methods and Table 3). Precursor types are defined by their adult products: pEC (only ECs), pEC/FSC (ECs and FSCs, with or without FCs), pFSC (FSCs with or without FCs), and pFC (only FCs).

The online version of this article includes the following figure supplement(s) for figure 4:

**Source data 1.** Numerical data for graphs in *Figure 4*.

ECs plus FCs was roughly compatible with a simplified model where 30% of labeled ovarioles derive from a single marked cell and 70% derive from exactly two marked cells. We then used that model to infer single-cell lineage frequencies and constituent cell numbers that would give rise to the scored raw data (see Materials and methods). From those numbers, we deduced the number of dividing precursors at pupariation to be about 24 in total, with 15.1 precursors giving rise only to ECs, 3.1

**Table 3.** Clone frequencies and cellular yields from different lineage experiments together with deduced single-lineage parameters including the number of precursors of different types.

| | N | EC | | Ec/FSC | | | FSC | | FC | EC + FC | | Number of dividing precursors per germarium | | | | |
| | | Freq. | Yield | Freq. | Yield | | Freq. | Yield | Freq. | Freq. | Yield | Total (n) | pEC | pEC/ FSC | pFSC | pFC |
| | | % | EC | % | EC | FSC | % | FSC | % | % | EC | 16/ (qx + ry) | pn | qn | rn | sn |
| | | 100 p | | 100q | | x | 100 r | y | 100 s | | | | | | | |
| 0 h APF MARCM raw data | 98 | 46.9 | 2.4 | 24.5 | 4 | 4 | 3 | 4.3 | 9.2 | 16.3 | 2.9 | | | | | |
| 0 h APF MARCM assuming 70 % double clones | | 63 | 1.8 | 13 | 3.2 | 3.5 | 5 | 4.3 | 19 | | | 24 | 15.1 | 3.1 | 1.2 | 4.6 |
| 0 h APF Mulicolor raw data | 26 | 42.3 | 3 | 30.7 | 3.5 | 2.6 | 7.7 | 4 | 11.5 | 7.7 | 2.5 | | | | | |
| 0 h APF Multicolor assuming 30 % double clones | | 52 | 2.4 | 24.5 | 2.1 | 2.3 | 9.5 | 3.9 | 14 | | | 17.3 | 9 | 4.2 | 1.6 | 2.4 |
| 36 h APF MARCM raw data | 121 | 20.7 | 2.5 | 23.1 | 2.8 | 3.5 | 3.3 | 3.3 | 9.1 | 43.8 | 2.6 | | | | | |
| 36 h APF MARCM assuming all double clones | | 51 | 1.8 | 7 | 3 | 4.7 | 4.1 | 3.1 | 38 | | | 34.9 | 17.8 | 2.4 | 1.4 | 13.3 |
| bond-GAL4 raw data | 73 | 38.4 | 5.1 | 48 | 8.3 | 5.4 | 1.4 | 7 | 2.7 | 9.6 | 9 | | | | | |
| bond-GAL4 assuming all double clones | | 61.9 | 3.3 | 24 | 5.6 | 4 | 6.3 | 5.3 | 7.8 | | | 15.1 | 9.4 | 3.6 | 1 | 1.2 |

APF: after puparium formation; EC: escort cell; FC: follicle cell; FSC: follicle stem cell.

precursors producing ECs and FSCs, 1.2 precursors producing FSCs but no ECs, and 4.6 precursors producing only FCs on average (*Figure 4E*, *Table 3*).

Similar modeling and calculations were performed for multicolor GFP-only clones (see Materials and methods). We deduced that a single precursor cell gives rise to both ECs and FSCs in 25% of all marked lineages and that the frequency of such precursors greatly exceeds precursors that produce FSCs but no ECs (10%). The deduced numbers of precursors of each type (9.0 EC, 4.2 EC/FSC, 1.6 FSC, 2.4 FC, and 17 total precursors) are reasonably similar to those calculated for the similarly timed MARCM experiment (*Figure 4E*, *Table 3*). Differences may result from variations in the exact developmental stage of animals of different genotypes at the time of clone induction.

We also performed a MARCM analysis of clones induced 3.5d prior to eclosion (36 hr APF) and examined in 0d adults to learn more about the timing of FC specification (see below). In this experiment, the proportion of ovarioles that were labeled (94%) and the frequency of EC/FC clones were particularly high, probably because there are many more dividing precursors than at 0 hr APF, so the deduction of single-cell lineage content is likely less reliable. Nevertheless, we estimated (see Materials and methods) that the proportions of single-cell lineage types were EC-only (51%), EC/FSC(±FC) (7%), FSC(±FC) (4%), FC-only (38%), and that the total number of dividing precursors was about 35 (*Figure 4E*, *Table 3*). Based on earlier analysis of MARCM lineages induced at –4d and –3d and examined at +2d , there should also be between 5 and 18 non-dividing EC precursors at –3.5d (*Table 2*). Thus, we estimate that at 36 hr APF there are 13.3 FC-only precursors within a total precursor population of about 40–53 (35 dividing plus 5–18 non-dividing cells) (*Table 3*). The most striking difference between lineages induced at 36 hr versus 0 hr APF is a large increase in lineages producing only FCs. Later, it will be shown that this corresponds to an expansion of FC precursors that accumulate posterior to germline cells during this period.

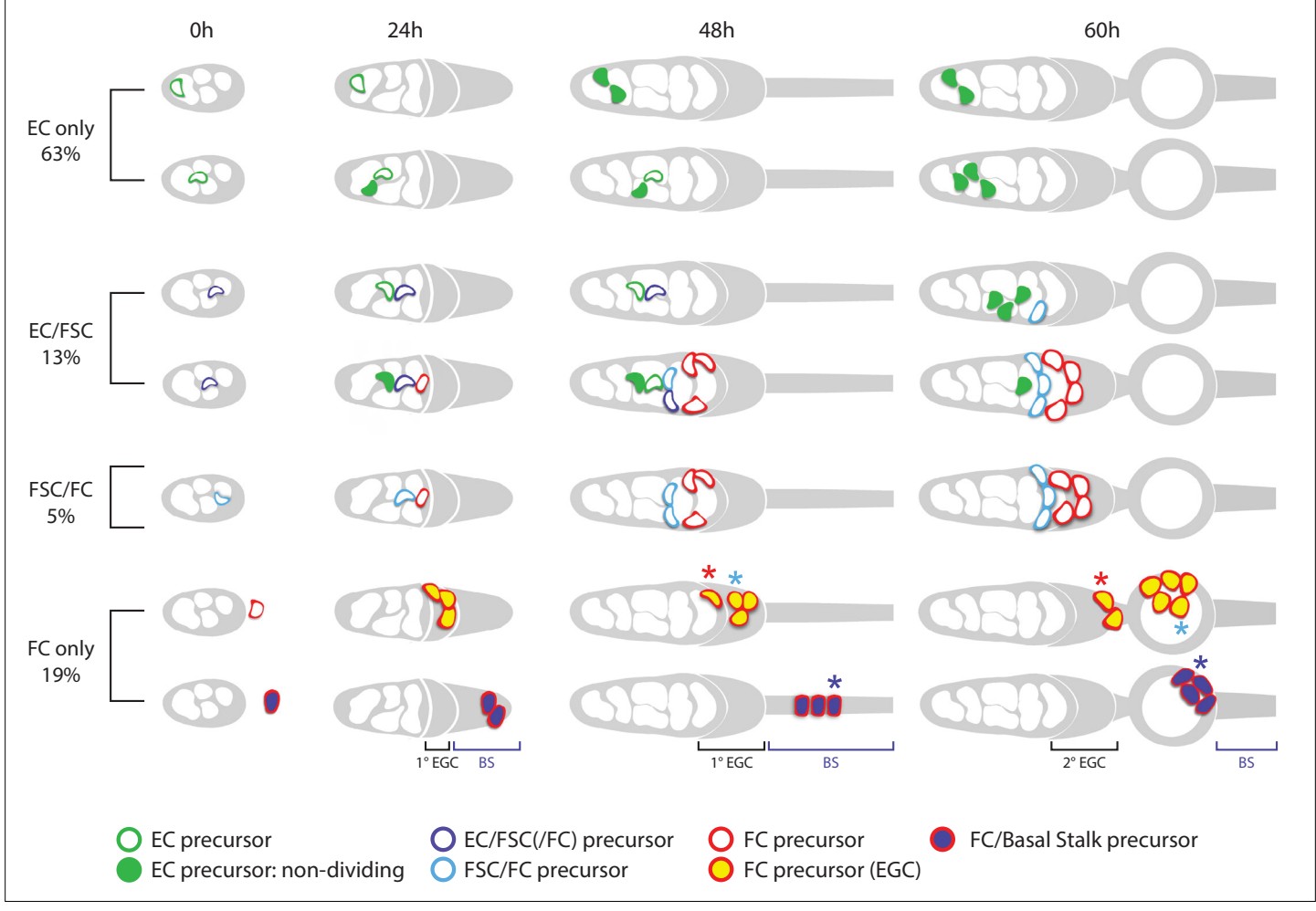

**Figure 5.** Summary of precursor lineage development. Summary cartoon of the fate of single precursor types from 0 hr after puparium formation (APF) until 60 hr APF deduced from lineage studies and pupal ovary imaging. The indicated frequency of clone types is drawn from the inference of single-cell lineages for MARCM clones induced at 0 hr APF (Table 3). Most 0 hr APF precursors generate only adult escort cells (ECs), with more anterior (left) precursors producing only r1 ECs (top row) and more posterior precursors also producing r2a ECs and more cells in total (second row). Filled green cells have ceased division. Most 0 hr APF follicle stem cell (FSC)-producing precursors also produce ECs (purple outlines). Some of these, more anterior, precursors (third row) do not produce follicle cells (FCs) but most (fourth row) also produce cells (red outline) that will become FCs (but not in the first-formed egg chamber). Precursors that produce adult FSCs but not ECs (blue outline) also produce FCs (fifth row). The most posterior precursors produce only FCs. A subset of progeny accumulate posterior to the germarium from 24 hr APF onward in the primary (1°) EGC (depicted as a posterior gray crown) and further posterior in the basal stalk (BS). Examples of origination from extra-germarial crown (EGC) cells (yellow fill, row 6) and BS cells (purple fill, row 7) at 24 hr APF are shown, but the most posterior intermingled cells (ICs) (just anterior to EGC cells) can also yield lineages containing only FCs. Most cells in the EGC will become FCs of the first budded egg chamber (sixth row; the cell denoted by the blue asterisk is modeled after the cyan cell in *Figure 9*). Others, like the cell on the posterior cyst at 48 hr APF, denoted by the red asterisk (modeled on the top red cell in *Figure 9*), move into the secondary EGC at 60 hr and later become FCs on the second budded egg chamber. Most precursors present in the BS at 48 hr APF (row 7) move onto the first budded cyst (the cell denoted by the blue asterisk is modeled after the white cell in *Figure 9*). Some of the most posterior BS precursor progeny remain in the BS throughout pupation (not shown). More posterior precursors are shown producing larger numbers of progeny because anterior cells divide more slowly and arrest division prior to eclosion.

## Timing of FC recruitment to first egg chamber

To determine when FCs are first allocated to the most mature germline cyst, we looked at the distribution of marked cells in 'FC-only' clones (ovarioles with marked FCs but no marked FSCs). The majority of newly eclosed adults have four egg chambers, and we excluded all exceptions from the analysis described below. For FC-only clones induced at 36 hr APF, more than half (34 out of 49) included labeled FCs in the fourth (most mature) egg chamber. Two-thirds of those (21 out of 34) had no other labeled FCs, showing that many precursors are restricted by 36 hr APF to contribute only to the

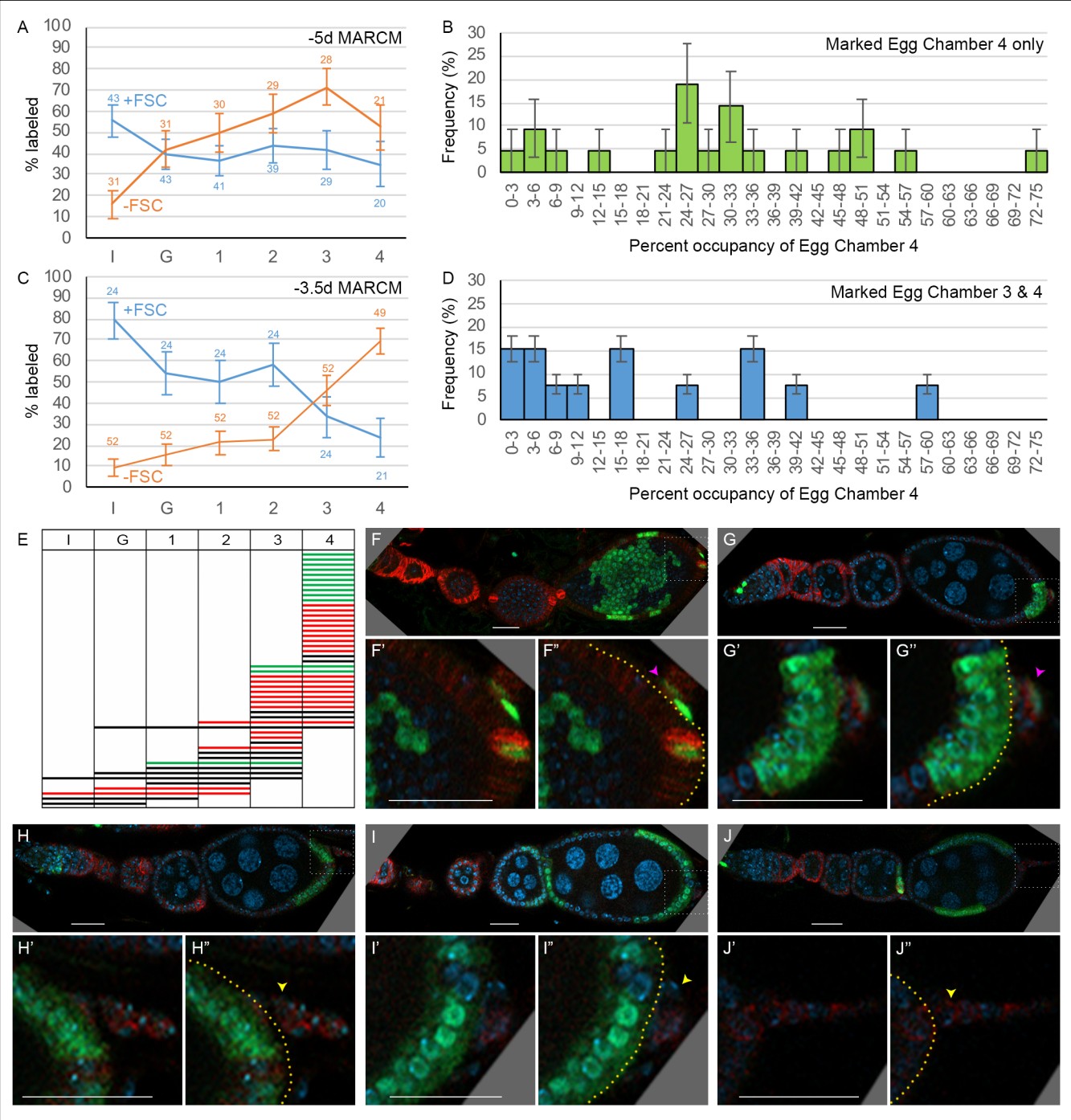

**Figure 6.** Patterns of follicle cell (FC) contributions indicate distinct pools of precursors along the AP axis. (**A, C**) Percentage of ovarioles containing labeled FCs that have marked FCs in germarial region 2b (I), region 3 (G), or egg chambers 1, 2, 3, or 4 (see *Figure 2A*) in newly eclosed adults for (**A**) −5d MARCM and (**C**) −3.5d MARCM clones with follicle stem cells (FSCs) (blue line) or with no FSCs ('FC-only,' orange line). The total number of ovarioles scored is indicated either side of the error bars (SEM); some ovarioles had only three egg chambers and some were not fully imaged. (**B, D**) MARCM clones induced at −3.5d and analyzed in newly eclosed adults. Marked occupancy of egg chambers was scored as a percentage of all FCs in egg chamber 4 that were marked when there were marked FCs (**B**) only in egg chamber 4 or (**D**) in egg chambers 3 and 4. The frequency of different occupancy ranges is displayed in bins of 3% . (**E**) Distribution amongst germarial region 2b (I), region 3 (G), and egg chambers 1–4 of FC-only clones (no marked FSCs) in newly eclosed flies after marking at −3.5d. Each row indicates a single ovariole. The colors indicate if a marked cell is present in the basal stalk (green), absent from the basal stalk (red), or if the basal stalk was not clearly imaged (black). (**F–J**) Ovarioles of newly eclosed flies with MARCM recombination induced at −3.5d containing FC-only clones (GFP: green; Fas3: red; DAPI: blue). Marked FCs can be seen in (**F, G**) the basal stalk (pink arrowheads) and egg chamber 4, (**H**) egg chamber 4 only, or (**I, J**) in egg chambers 4 and 3 only (yellow arrowheads indicate Fas3-positive basal

*Figure 6 continued on next page*

*Figure 6 continued*

stalk with no GFP-positive FCs). Insets show enlargements of boxed regions; yellow borders outline the posterior edge of egg chamber 4, with cells further posterior located in the basal stalk. Scale bars, 20 µm.

The online version of this article includes the following figure supplement(s) for figure 6:

**Source data 1.** Numerical data for graphs in *Figure 6A*.

**Source data 2.** Numerical data for graphs in *Figure 6C*.

**Source data 3.** Numerical data for graphs in *Figure 6B and D*.

first-formed egg chamber (*Figure 6E and F*). The 13 exceptions all had marked FCs in egg chamber 3 (*Figure 6E, I, J*). This frequency is too high for all to be explained by the presence of two lineages. We also found that the proportion of the egg chamber 4 epithelium occupied by marked FCs was significantly lower when marked FCs were also in egg chamber 3 (19.6%) than when confined to only the terminal egg chamber (29.7%) (*Figure 6B and D*). These results suggest that, in most cases, a precursor labeled at 36 hr APF divided before all egg chamber 4 FCs were allocated. In some cases, one daughter became an FC founder for egg chamber 4 and the other daughter contributed to egg chamber 3. In other cases, both daughters contributed to egg chamber 4, producing a higher average FC contribution for that labeling pattern. We do not know the cell cycle time for these FC-only precursors subsequent to 36 hr APF, but we expect (from amplification data for EC/FSC lineages in *Table 1*) that it is shorter than 24 hr and may be as short as 11 hr, the cycling time for early FCs in adult ovarioles (*Margolis and Spradling, 1995*), so we estimate that FCs are allocated to egg chamber 4 at about 47–60 hr APF. This estimate fits with direct morphological observations described later.

By 48 hr APF, all germaria have a single cyst in the most posterior region (*Figure 1—figure supplement 1*) but neighboring somatic cells (ICs) only express low levels of Fas3 (*Figure 1H* and *Figure 7—figure supplement 1A*). In adult germaria, even the most recently recruited FCs surrounding the penultimate germarial cyst express high levels of Fas3 (*Figure 1A*); a subset of those cells must then become specialized polar and stalk cells before the egg chamber can bud about 24 hr later (*Figure 7—figure supplement 1D*). The most posterior ICs that do not express Fas3 strongly before 48 hr APF therefore appear to be far behind schedule, given that the first egg chamber will bud at about 56 hr APF on average (*Figure 1I*). These observations suggest that ICs may not be the cells that become the FCs of the first budded egg chamber, as might have been assumed by analogy to budding of adult egg chambers.

## Lineage evidence for separate precursors of FCs in the first-formed egg chamber

The distribution of marked FCs in lineages initiated at 36 hr APF (*Figure 6E*) suggests that precursors of the first-formed egg chamber contribute only to the two terminal egg chambers (seen in 32/34 cases). The two exceptions (*Figure 6E*) likely contain more than one lineage. Five ovarioles had marked FCs only anterior to the two terminal egg chambers, while seven included marked FCs in egg chamber 3 and more anterior locations. It seems unlikely that the latter all include two lineages; all seven have marked FCs also in egg chamber 2, compatible with a common origin. Thus, by 36 hr APF the FC-only precursors appear to be segregated into two populations. The majority contribute FCs to the first-formed egg chamber and sometimes also to egg chamber 3, while the remainder contribute FCs to more anterior egg chambers, sometimes including egg chamber 3.

Ovarioles that included marked FSCs only rarely had marked FCs in the terminal egg chamber (*Figure 6C*). The five exceptions (out of 21) may derive from ovarioles that harbor an FC-only lineage (53% frequency overall) in addition to an FSC-containing lineage (26% overall). Thus, at 36 hr APF it appears that more than two-thirds of FC-only precursors will contribute to the first-formed egg chamber (with some also contributing to egg chamber 3), while more anterior egg chambers are later populated by derivatives of a combination of FC-only, FSC/FC, and EC/FSC/FC precursors.

## An extra-germarial crown (EGC) forms prior to egg chamber budding

Posterior to the interspersed germline cells and somatic ICs of the pupal germarium are somatic cells with strong Fas3 staining from 24 hr APF onward (*Figures 1 and 7*). We noticed that the most anterior of these Fas3-positive cells express TJ, which is also expressed in ICs and required for their

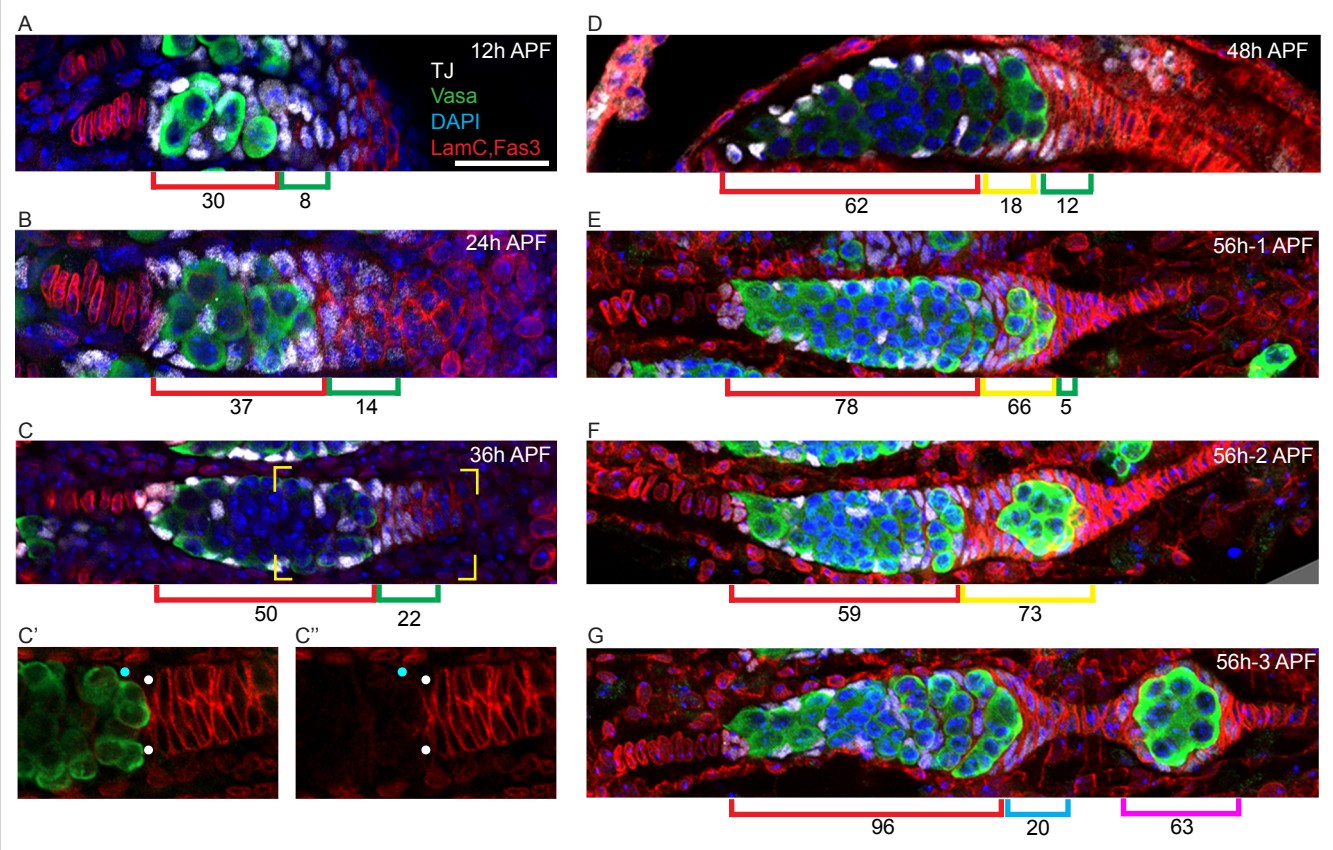

**Figure 7.** Extra-germarial crown cells (EGCs); a potential source of early follicle cells (FCs). (A–G) Traffic Jam (TJ)-positive cells (white) were counted in ovaries stained also for Vasa (green), Lamin-C and Fas3 (both in red), and DAPI (blue). Bar, 20 μm applies to all images. (A–C) Up to 36 hr after puparium formation (APF), prior to budding of the first egg chamber, intermingled cells (ICs) (red brackets), defined as intermingled with germline cells (green) in the developing germarium, expressed TJ (white) but little or no detectable Fas3 (red). Cells posterior to the developing germaria expressed Fas3 and were previously all termed basal cells and expected to intercalate to form the basal stalk. However, a subset of Fas3-positive cells immediately posterior to the germarium also expressed TJ. Those cells formed an EGC of increasing cell numbers, indicated by green brackets. (C', C") At 36 hr APF, some somatic cells contacting the posterior of the most advanced cyst expressed Fas3 strongly (white dots) and some did not (blue dots). Fas3-positive cells were counted as EGC cells and Fas3-negative cells as ICs. (D) At 48 hr APF, the most posterior germline cyst is about to leave the germarium in most samples. Here, the cyst (green) appears to have just started to leave, invading Fas3-positive cells in EGC territory. The Fas3-positive cells surrounding the cyst are indicated by a yellow bracket, while Fas3-positive and TJ-positive cells posterior to the cyst are indicated by green brackets. Average IC (red brackets) and EGC (green and yellow brackets) cell numbers are written underneath the brackets and were derived from counting (A) 5, (B) 5, (C) 10, and (D) 5 samples. (E–G) Three developing ovarioles from the same 56 hr APF ovary, representing progressively later stages of development from top to bottom, show the most posterior cyst leaving the germarium and surrounded by Fas3-positive, TJ-positive cells. (E, F) Fas3-positive, TJ-positive cells posterior to the germline (green bracket), and around the most posterior cyst (yellow bracket) were counted for these single samples and then (G) counted as on the budded cyst (magenta bracket) or anterior to the budded cyst and forming a secondary EGC posterior to the germarium (blue bracket). Those two populations are separated by stalk cells that do not stain for TJ. The Fas3 patterns in 48 hr, 56 hr, and 60 hr APF and adult ovaries are compared in *Figure 7—figure supplement 1*. See also *Figure 1—figure supplement 1*.

The online version of this article includes the following figure supplement(s) for figure 7:

**Figure supplement 1.** Somatic cells surrounding the most posterior germarial cyst do not show strong Fas3 expression prior to budding.

interspersion with germline cells (*Li et al., 2003*). These TJ/Fas3-positive cells form a small multi-layered cone or crown around the posterior of the developing germarium that increases in size and cellular content from 12 to 36 hr APF (*Figure 7A–C*). We refer to those TJ/Fas3-positive cells beyond the most posterior germline cyst as the EGC, and to the more posterior Fas3-positive cells with no TJ expression as basal stalk cells. Expression of Fas3 distinguishes adult FCs from other TJ-expressing cells (FSCs and ECs) (*Figure 7—figure supplement 1D*; *Reilein et al., 2017*), suggesting that EGC cells may become the first FCs.

We also noticed that after budding the first egg chamber was often found much further from the germarium than seen in adults, where a short TJ-negative stalk separates the two (*Figure 1A and F*). The intervening space was occupied by a partially intercalated stalk of TJ-negative cells and by a small 'secondary' EGC of Fas3- and TJ-positive cells, crowning the germarium (*Figure 7G*). After one cyst had budded and a second cyst had rounded up in preparation for budding (around 60 hr APF), strong Fas3 expression was also seen in germarial somatic cells (*Figure 7—figure supplement 1C*), resembling the pattern seen in adults (*Figure 7—figure supplement 1D*). In samples with two budded egg chambers (72–84 hr APF) or more, the newly budded egg chamber is close to the germarium (*Figure 1J*). Thus, formation of the first egg chamber appears to involve a morphological process quite different from adult egg chamber budding; this is partly recapitulated by the second egg chamber before adopting a typical adult morphology.

## Live imaging prior to egg chamber budding; germline cysts move past IC cells

We used a live imaging protocol that we had previously developed for adult ovarioles (*Reilein et al., 2018a*) in order to view pupal ovaries and track individual somatic cells. We were able to follow isolated live ovaries for up to 15 hr, during which time many cell movements and cell divisions were observed, suggestive of continued normal behavior. We first used samples where all cells were labeled by expression of a *ubi-GFP* transgene. Later, we used animals with multicolor clones induced 2–3d earlier, so that some cells had lost GFP or RFP expression or both (*Figure 3A*). Variability in the levels of GFP and RFP simplified the task of following individual cells over time. We were able to track the movements of multiple cells in a series of videos initiated at a variety of times after pupariation.

From about 30 hr APF onward, tracked ICs moved short distances relative to other cells in a variety of directions. Most ICs maintained their relative AP locations within the developing germarium as cysts moved past them. For example, one of two initially parallel germline cysts adopted the most posterior position in the germarium while adjacent somatic ICs remained in position (*Figure 8A and B*). Similarly, as a cyst moved posterior to contact EGCs (magenta arrow, *Figure 8C*), overlying and posterior ICs ended up anterior to the cyst (white, yellow, and red arrows, *Figure 8C*). These observations support lineage analysis deductions of limited, stochastic precursor movements along the AP axis and suggest that 16 cell germline cysts do not have a stable coating of somatic cells from about 30–50 hr APF. During the 30–50 hr APF period, there were cell divisions in the EGC and stalk (*Figure 8—video 1*) but we saw no evidence of cells moving from the IC region into the EGC or in the reverse direction (see outlined red EGC cells in *Figure 8B*), suggesting that the EGC population grows primarily as a result of EGC cell division.

## Live imaging shows EGC and basal stalk cells become FCs of the first egg chamber

Imaging of an ovary from about 48 hr APF onward captured the key process of egg chamber budding (*Figure 9*, *Figure 9—figure supplements 1–3*, *Figure 9—videos 1–3*). At 48 hr APF, fixed images showed that cells surrounding the posterior half of the germline cyst express Fas3 (*Figure 7D*, yellow bracket), consistent with evidence from live imaging that the cyst is already moving into EGC territory (*Figure 8C*), as described above. The location of Fas3-positive TJ-positive EGC cells around (yellow brackets, *Figure 9*) and beyond (green brackets, *Figure 9*) the germline cyst is inferred by comparison to similarly staged fixed images (*Figure 7*), with Fas3-positive TJ-negative cells posterior to the EGC (beyond the yellow and green brackets) considered basal stalk cells.

The most posterior germline cyst (highlighted by green nuclei in the lower germarium of *Figure 9*) moved out of the germarium into EGC territory as EGC cells enveloped the cyst. For example, the cyan-outlined cell started within the EGC, contacted the cyst prior to 8.3 hr and remained associated for the next 6.5 hr (*Figure 9*, *Figure 9—figure supplement 2*). Similarly, the pink cell, just posterior to the cyst at the start of imaging, moved to the midpoint of the budded egg chamber (*Figure 9*, *Figure 9—figure supplement 1C*, *Figure 9—video 1*).

The cells outlined in orange and green, like the yellow-outlined cell in the germarium above, started close to the anterior face of the terminal cyst. These cells were likely the most anterior EGC cells prior to budding (a few hours prior to imaging). They all lost contact with the cyst before 7 hr and were posterior to the second cyst in the secondary EGC by the end of imaging (*Figure 9*, *Figure 9—figure*

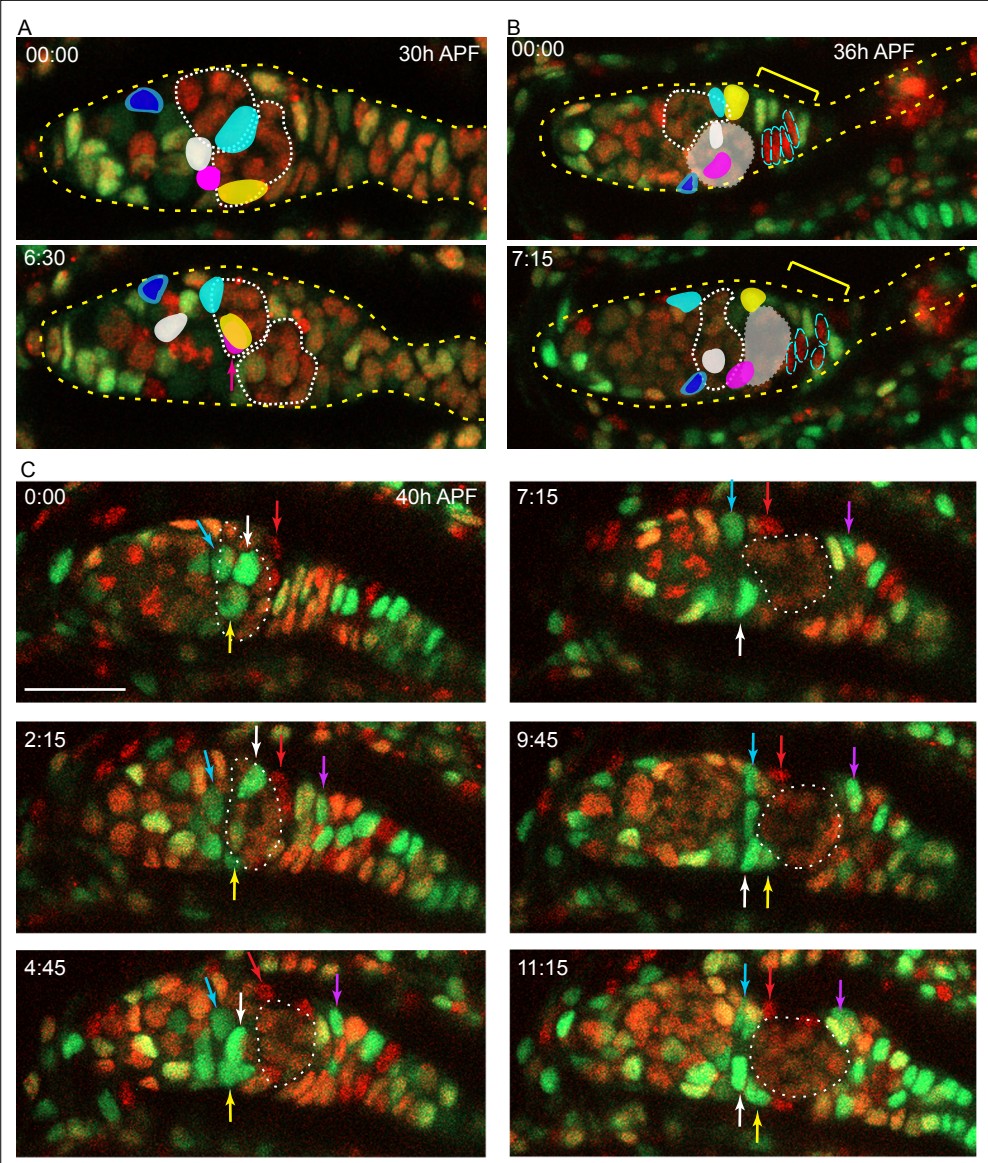

**Figure 8.** Intermingled cells (ICs) move independently of germline cysts and do not mix with extra-germarial crowns (EGCs). Live imaging was performed on multicolor pupal ovaries with clones induced by heat-shock such that cells could lose RFP, GFP, or both. (**A**) Z-projections of 0 hr and 6 hr 30 min timepoints of a 30 hr after puparium formation (APF) germarium showing five tracked ICs (colored). ICs were located at the outer surface of the germarium and moved independently of one another in posterior, anterior, and lateral directions (around the circumference). The most posterior germline cyst (outlined by a white dotted line) moved posteriorly past the cyan and yellow IC cells. The relative disposition of ICs and cysts changed considerably, suggesting that they are not strongly associated. Only z-planes containing tracked cells were included in the projection (z4, 5, 6, 7, and 8 at 00:00 and z5, 6, 9, and 10 at 6:30). (**B**) Z-projections of 0 hr and 7 hr 15 min timepoints of a 36 hr APF germarium showing five tracked ICs. ICs moved independently of one another and remained in their domain as the most posterior germline cyst (highlighted by shading) moved past. Four cells in the EGC (outlined red cells) remained in the EGC (indicated by the yellow bracket). Projected sections are z5–9 at 00:00 and z4, 7, 8, and 9 at 7:15. Individual z-sections of the germaria in (**A**) and (**B**) are shown in *Figure 8—figure supplement 1*. (**C**) Z-projections of selected timepoints of a 40 hr APF germarium imaged for 11 hr 15 min, showing three tracked green ICs (white, yellow, and blue arrows); ICs remained in their domain as the posterior germline cyst (outlined by a white dotted line) moved past them. Four red cells (one is shown by the red arrow) started on the posterior of the posterior cyst and moved anterior along the cyst; cells in a neighboring no-color clone moved similarly. A green cell that began in the EGC (magenta arrow) moved onto the posterior edge of the cyst. The accompanying *Figure 8—video 1* of this germarium shows mitotic cells in the EGC and the stalk. Bar, 20 μm. See also *Figure 8—figure supplement 1*.

*Figure 8 continued on next page*

*Figure 8 continued*

The online version of this article includes the following video and figure supplement(s) for figure 8:

**Figure supplement 1.** Intermingled cells (ICs) move independently of one another and do not mix with extra-germarial crowns (EGCs).

**Figure 8—video 1.** A 40 hr after puparium formation (APF) multicolor germarium imaged for 11 hr 15 min.
https://elifesciences.org/articles/69749/figures#fig8video1

*supplement 1*, *Figure 9—figure supplement 5B*). The red-outlined cell (red asterisk), which started in a more posterior location, also ended in the secondary EGC (*Figure 9*, *Figure 9—video 2*). Thus, during budding, more posterior EGC cells become associated with the germline cyst exiting the germarium, while more anterior EGCs move past the cyst into the secondary EGC, awaiting budding of the next germline cyst.

Surprisingly, we observed that even distant basal stalk cells migrated onto the budding germline cyst. The white-outlined cell, which initially lay posterior to the region with TJ expression (from comparison to fixed images), entered EGC territory within 4 hr and became associated with the germline cyst by 12 hr (*Figure 9*, *Figure 9—figure supplement 2*, *Figure 9—video 3*). Thus, over the nearly 15 hr of live imaging the newly budded cyst acquired a set of closely associated cells of epithelial appearance that derived from the EGC and basal stalk cells.

Live imaging of a different ovariole from 55 hr APF onward showed continued posterior movement of the budding cyst until only a small number of basal stalk cells remained at its posterior (*Figure 9—figure supplement 4B* and *Figure 9—video 4*). During this almost 15 hr viewing period, the number of cells posterior to the cyst declined from 32 to 13 (*Figure 9—figure supplement 4C*). Fixed images of later stages, after two egg chambers have budded (84 hr APF; *Figure 9—figure supplement 4D*), revealed an average of nine cells in the stalk posterior to the terminal egg chamber, resembling the structure seen in newly eclosed adults (*Figure 2A*). Prior to budding, there were 50–65 cells posterior to the germline throughout the time range of 24–48 hr APF (*Figure 9—figure supplement 4A*).

Imaging from about 60 hr APF onward revealed the most posterior cyst in the germarium rounding up and moving a short distance into the secondary EGC, which includes dividing cells (*Figure 9—figure supplement 5A–C*). Tracked cells on the first budded egg chamber moved short distances anterior as the stalk elongated between the first and second egg chambers (*Figure 9—figure supplement 5C*). ICs anterior to the second budding cyst moved posterior as the germarium elongated and appeared poised to associate with later cysts (*Figure 9—figure supplement 5A*). We infer that the second budded egg chamber has FCs derived from the secondary EGC, which is itself derived from primary EGC cells and perhaps also some posterior ICs at 48 hr APF. FCs on later egg chambers likely derive solely from cells that were ICs at 48 hr APF.

## Lineage analysis of precursor contributions to the first-formed FCs and the adult basal stalk

Live imaging indicated that many cells in the basal stalk at 48 hr APF later become FCs on the first budded egg chamber, while a small number do not migrate over the cyst and form a short basal stalk structure of 5–10 single-file cells posterior to egg chamber 4 in newly eclosed adults (*Figure 2A*). Consistent with this scenario, when we examined lineages marked at 36 hr APF, we found 13 ovarioles with label in the basal stalk and in FCs of egg chamber 4 (*Figure 6F and G*), suggesting a common origin from basal stalk cells at 36 hr APF. There were 18 ovarioles where FCs in egg chamber 4 were marked but the adult basal stalk was not, consistent with an origin from EGC or anterior basal stalk cells at 36 hr APF (*Figure 6E and H–J*). The proportion of ovarioles containing marked FCs in egg chamber 4 that also contained marked FCs in egg chamber 3 was higher when marked basal stalk cells were absent (7/18) than when they were present (2/13), consistent with a more anterior origin of lineages that did not include the adult basal stalk (*Figure 6E*).

In summary, live imaging, together with lineage analysis initiated at 36 hr APF, revealed a separate group of FC-only precursors, now identified as basal stalk and EGC cells posterior to the germarium, that become FCs of the first-formed egg chamber and contribute also to the second-formed egg chamber. Lineage analysis revealed additional FC-only precursors that contribute to more anterior FCs. These FCs must be formed within the germarium (from ICs) because the germarium has an adult

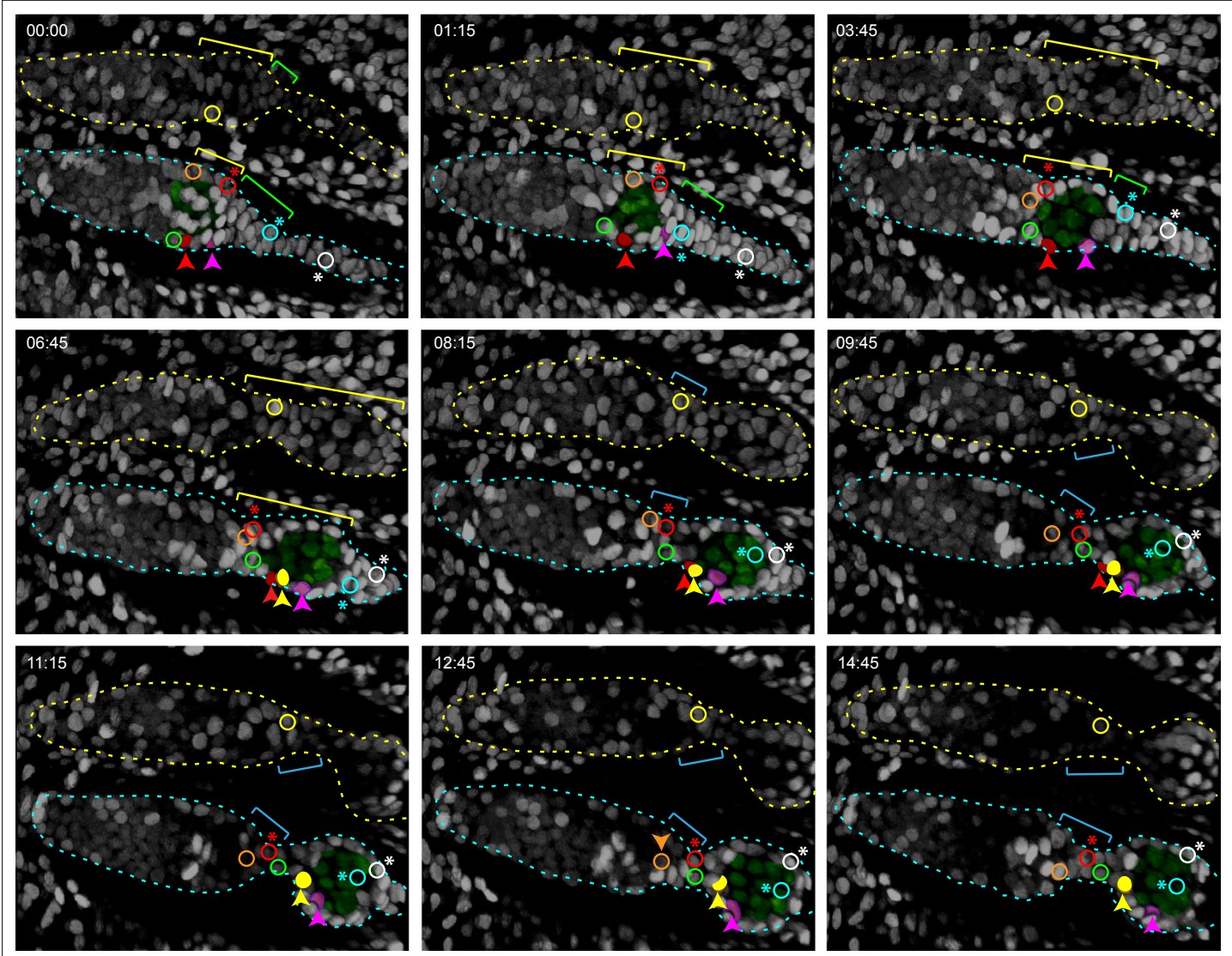

**Figure 9.** Budding of the first egg chamber: extra-germarial crown (EGC) and basal stalk cells become follicle cells (FCs). Time-stamped frames from a 3D reconstruction of a movie of 14 hr 45 min starting at 48 hr after puparium formation (APF) and showing budding of an egg chamber. All cells are labeled white from a *ubi-GFP* transgene. Three cells that were visible near the surface of a 3D reconstruction are colored in solid red, pink, and yellow in the lower germarium, together with five other cells (open circles) that were in interior z slices. All marked cells were tracked using Zeiss Zen software for 60 timepoints over 14 hr 45 min, except for the red cell that could not be reliably tracked after 10 hr and the yellow cell in the lower germarium, which was not tracked before 5 hr 45 min. The posterior germline cyst in the lower germarium is highlighted in green. Cyst cells were distinguishable from somatic cells because the nuclei were larger and paler green. The yellow brackets indicate Fas3+ TJ+ cells, inferred from comparison to fixed images at the same stages. The cyst slid past cells located at the anterior edge of the cyst which were inferred from comparison to fixed 48 hr APF images to be Fas3-positive and to be the most anterior EGC cells when the cyst first started to move out of the germarium (just prior to 48 hr APF, see *Figure 8*). These cells were later either at the posterior edge of the germarium (orange-outlined cell and yellow-outlined cell in the upper germarium) or between the germarium and the budded cyst (green-outlined cell) in the nascent secondary EGC (blue brackets), as seen more clearly in *Figure 9—figure supplement 1A,B* (yellow-outlined cell) and *Figure 9—figure supplement 3A,B* (orange and green-outlined cells). The cell marked with the open red circle (asterisk; modeled in *Figure 5*). *Figure 8* began on the posterior cyst and ended in the secondary EGC (*Figure 9—video 2*). The cyan cell (asterisk; modeled in *Figure 5*) initially towards the posterior of the primary EGC (indicated by green brackets) adopted progressively more anterior positions on the germline cyst after the cyst moved into EGC territory. The white cell (asterisk; modeled in *Figure 5*) was originally in a location posterior to the EGC (it is expected, from comparison to fixed images, not to express Traffic Jam [TJ]) but then moved into EGC territory by 3 hr 45 min and ended up close to the cyan cell on the budded germline cyst (*Figure 9—video 3*). The locations of the cyan and white cells covering the posterior of the cyst can be seen clearly in *Figure 9—figure supplement 2*. Individual z-sections showing the locations of the red, pink, and yellow cells in the lower germarium at the beginning, middle, and end of their imaging times are shown in *Figure 9—figure supplement 1* and *Figure 9—video 1*. Time is in hours:minutes. See also *Figure 9—figure supplements 1–5*.

*Figure 9 continued on next page*

*Figure 9 continued*

The online version of this article includes the following video and figure supplement(s) for figure 9:

**Figure supplement 1.** Detail from live imaging (*Figure 9—video 1*) summarized in *Figure 9*.

**Figure supplement 2.** Further detail from live imaging (*Figure 9—videos 1; 3*) summarized in *Figure 9*.

**Figure supplement 3.** Further detail from live imaging (Video S2) summarized in *Figure 9*.

**Figure supplement 4.** Basal stalk and extra-germarial crown (EGC) cells are the source of the first follicle cells (FCs) and polar cells.

**Figure supplement 5.** Continued anterior movement of follicle cells (FCs) after budding of the first egg chamber.

**Figure 9—video 1.** A 48 hr after puparium formation (APF) germarium imaged for 14 hr 45 min.

https://elifesciences.org/articles/69749/figures#fig9video1

**Figure 9—video 2.** A 48 hr after puparium formation (APF) germarium imaged for 14 hr 45 min.

https://elifesciences.org/articles/69749/figures#fig9video2

**Figure 9—video 3.** A 48 hr after puparium formation (APF) germarium imaged for 14 h 45 min.

https://elifesciences.org/articles/69749/figures#fig9video3

**Figure 9—video 4.** To accompany Figure 9—figure supplement 4B.

https://elifesciences.org/articles/69749/figures#fig9video4

morphology, with no EGC structure, before the third egg chamber buds. We did not find any evidence for cells posterior to ICs entering the germarium at any stage during live imaging, consistent with all ICs present in the germarium after two cycles of budding originating from ICs present at 36 hr APF. Thus, the FC-only precursors that produce FCs anterior to egg chamber 3, and even more certainly, all precursors of FSCs, which are anterior to FC-only precursors, must reside within the germarium, as ICs, throughout pupal development.

## Pattern of precursor divisions along the AP axis

Progressive cessation of division from anterior to posterior over the EC-producing domain was deduced from the total number of r1 and r2a cells produced relative to FSCs after labeling dividing precursors with MARCM at different times, as described previously (*Table 2*). This inference was supported by examination of clone types and the number of cells they contained. The frequency of clones producing only ECs fell from 58% and 57% (0 hr and 24 hr APF) to 30% (48 hr APF) and then 14% (72 hr APF) (*Table 1*). Moreover, the yield of r1 ECs relative to r2a ECs declined from 2.2 (0 hr APF) and 2.6 (24 hr APF) to 0.9 (48 hr APF) and 0.8 (72 hr APF). The yield of marked cells per EC/FSC precursor was also consistently higher than for more anterior EC-only precursors at all times of clone induction, even though the many FCs produced by EC/FSC precursors were not included (*Table 1*). We also examined precursor cell cycling directly using EdU and FUCCI labeling.

We incubated dissected pupal ovaries at various developmental stages with EdU in vitro over a 1 hr time period to detect cells in S phase. From 0 to 24 hr APF, ICs throughout the germarium often incorporated EdU (*Figure 1—figure supplement 2A–C*). By 48 hr APF, the prevalence of anterior EdU-labeled ICs was quite low, whereas EGC cells had frequent EdU signals (*Figure 1—figure supplement 2D*). An occasional IC in the anterior half of the germarium could be found with EdU label at 60 hr APF but not at 96 hr APF (*Figure 1—figure supplement 2E and F*). The most anterior EdU signal at 96 hr APF in somatic cells was near the middle of the germarium, anterior to strong Fas3 expression, likely corresponding to r2a EC or EC/FSC precursors.

In the FLY-FUCCI system, RFP- and GFP-tagged proteins are produced throughout the cell cycle under the influence of a chosen promoter, while degrons from CycB and E2F1 promote rapid degradation of the RFP and GFP fusion proteins at the start of G1 and S phase, respectively (*Zielke et al., 2014*). We used *UAS*-driven *FLY-FUCCI* transgenes and found *C587-GAL4* to be more effective than *tj-GAL4* or *act-GAL4* in producing GFP and RFP FUCCI signals in developing and adult ovaries. In adult germaria, all r1 ECs express only GFP (*Figure 10G*), indicating G1 (or a G0 state). r2a ECs include a mix of GFP-only G1 cells and cells with both GFP and RFP, indicating G2 (*Figure 10G*). FSCs additionally include some cells with RFP-only (late S-phase) and no GFP or RFP (early S-phase, verified by co-labeling with EdU; D. Melamed pers. comm.).

From 12 to 36 hr APF, ICs at all AP locations in the germarium included some cells with each FUCCI color combination, although G1 cells were always most frequent in more anterior regions

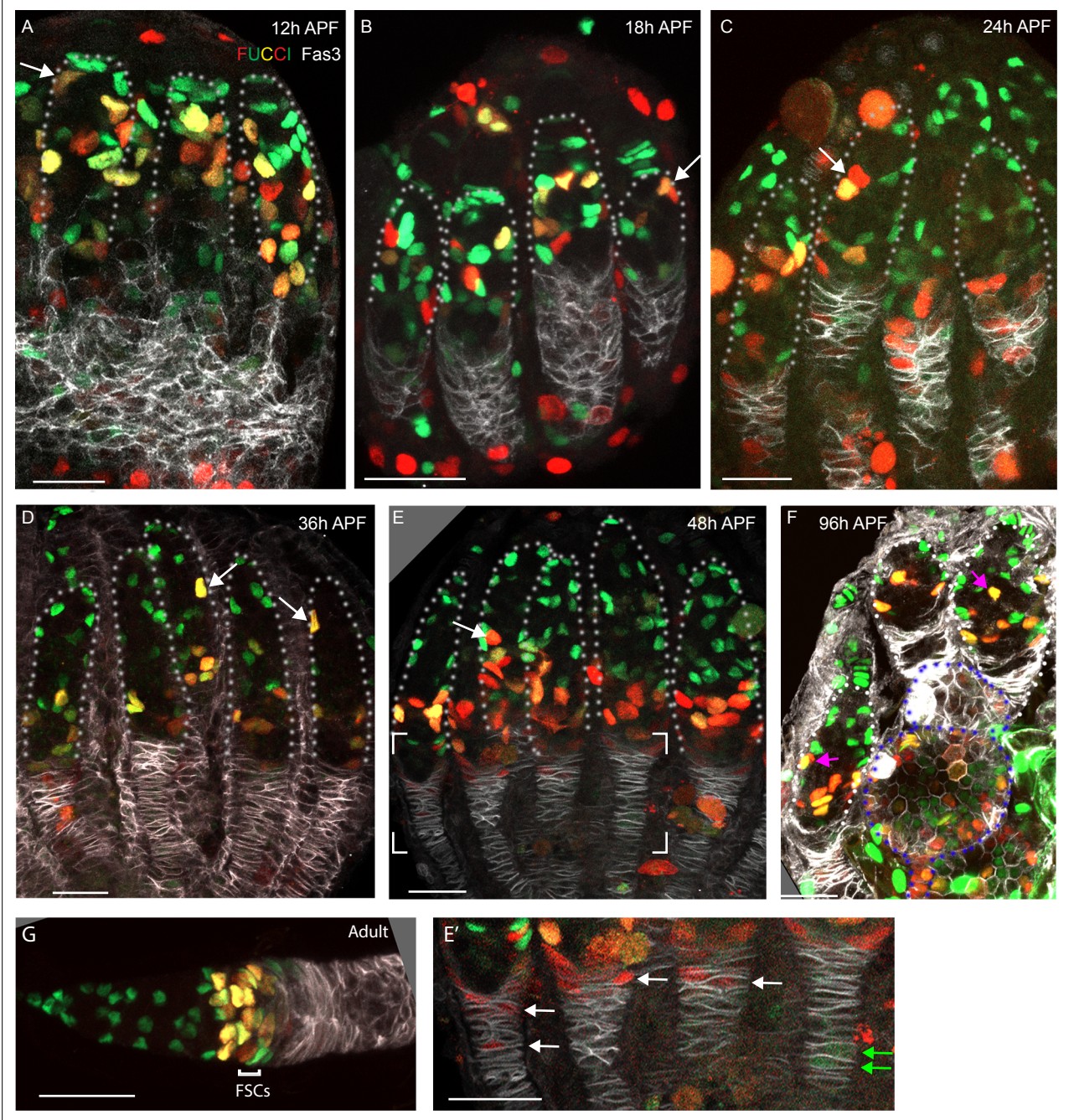

**Figure 10.** FUCCI labeling shows progressive accumulation of anterior intermingled cells (ICs) in G1 or a G0 arrest. (**A–G**) *C587-GAL4* was used to drive *UAS-FUCCI* expression and ovaries were stained for Fas3 (white) and Traffic Jam (TJ) (not shown) to identify all somatic cells. Somatic cells were either green (G1 or G0), red (late S phase), red/green (G2), or without any color (early S-phase for cells with adequate *C587-GAL4* expression). Germaria are outlined with dotted lines. All bars, 20 µm. (**A–F**) Many red or red/green cells are present in anterior locations close to cap cells at (**A**) 12 hr after puparium formation (APF) and (**B**) 18 hr APF (arrows) but (**C**) by 24 hr APF and (**D**) 36 hr APF fewer such cells are evident and are further from the anterior (arrows). (**E**) At 48 hr APF, ICs in the anterior third of the germarium are uniformly green (G1 or G0). The bracketed region is magnified with enhanced colors to reveal (**E'**) that many cells in the EGC are red (S-phase), while one or two more posterior, basal stalk cells are faintly green. (**F**) At 96 hr APF, after three egg chambers have budded (two are outlined with blue dotted lines), red/green ICs can still occasionally be seen in the anterior half of the germarium (pink arrows) but (**G**) in adults this territory, occupied by r1 escort cells (ECs), is entirely green, followed by a mix of green and red/green more posterior r2a ECs, and follicle stem cells (FSCs), which are predominantly red/green (G2), red or colorless (S-phase) and with only a few green (G1) cells. In adults, *C587-GAL4* expression extends through all ECs and FSCs but ends near the FSC/FC border.

(*Figure 10A–D*). By 48 hr APF, almost all somatic cells in the anterior half of the germarium were in G1 or G0 (GFP-only), consistent with substantially reduced cycling (*Figure 10E*), whereas a large fraction of EGC cells were in S phase (marked by RFP only; *Figure 10E'*). By 96 hr APF, after two or three egg chambers have budded, the germarial FUCCI pattern was similar to adults (*Figure 10F*). Thus, direct observation of EdU incorporation and cell cycle markers confirms the progressive decline in anterior IC proliferation over time that was inferred from lineage analyses and suggests that the anterior to posterior gradient of proliferation stretches all the way from the most anterior ICs to posterior ICs and EGC cells, which appear to cycle rapidly.

## Targeted lineage studies support inferences of a common precursor of FSCs and ECs

Recent single-cell RNA sequencing of developing ovaries in late third-instar larvae led to the discovery of a variety of genes with regional expression patterns, providing resources for targeted cell labeling (*Slaidina et al., 2020*). The authors identified a cell population posterior to ICs that preferentially expressed *bond* mRNA prior to pupariation. A *bond-GAL4* driver was then used together with a temperature-sensitive *GAL80* transgene to initiate GFP-marked lineages using standard G-trace reagents (*Evans et al., 2009*), keeping animals at 29C during larval stages to initiate labeling, switching to 18C (to terminate labeling) about 1 day after pupariation and scoring marked cells in 2d-old adults (*Slaidina et al., 2020*). Most of the resulting labeled ovarioles (31/32) were described as including marked FSCs or FCs. FSCs were inferred from the presence of marked FCs in the germarium and youngest egg chambers rather than being scored directly. Although more than half of the labeled ovarioles also contained marked ECs (19/32), the authors suggested that separate precursors gave rise to the marked ECs and to the marked FSCs plus FCs in the same ovariole, and that the FSC precursors reside posterior to ICs in late third-instar larvae. Both conclusions are directly contradicted by our single-cell lineage analyses and morphological studies. We therefore investigated lineages targeted by *bond-GAL4* using reagents donated by the authors. We scored results in newly eclosed adults in order to count FSCs directly and to include all FCs produced during pupation.

We found a high frequency of GFP-marked cells when animals were kept at 18C throughout (37 labeled ovarioles out of 76), suggesting 'leaky' clone initiation (we have observed that *tub-GAL80^ts* activity is insufficient to silence several *GAL4* lines during ovary development). Labeled ovarioles had marked ECs only or ECs together with FSCs present in almost all cases (35/37; *Figure 11A–D*). The proportion of ovarioles with marked ECs or FSCs was no higher when larvae were raised at 29C before being moved to 18C about 2d after pupariation (eclosing 6d later; 13 of 44 ovarioles labeled), or when animals were raised at 18C before shifting to 29C for the final 3d or 4d of pupation (23 of 88 ovarioles labeled), so the time of clone initiation cannot be inferred directly in any sample (*Figure 11D*).

The marked lineages were similar in their composition of cell types (*Figure 11A–D*) to those we described earlier using *hs-flp* to initiate clones at pupariation (5d before eclosion), suggesting that they may have been induced at a similar time. The majority of ovarioles from the *bond-GAL4* experiments (considering all three temperature conditions together) included either only marked ECs (about 40%) or marked ECs and FSCs (about 50%, almost all together with marked FCs) (*Figure 11D*). Only 1 out of 73 labeled ovarioles had marked FSCs (and FCs) but no marked ECs. These results clearly provide no direct evidence of the labeling by *bond-GAL4* of precursors that produce FSCs and FCs without also producing ECs.

The number of labeled ECs and FSCs (when present) per germarium was consistently higher than we observed with *hs-flp* induced MARCM or multicolor clones induced at pupariation (*Figure 11A–C*, *Table 3*), suggesting that most labeled ovarioles in the *bond-GAL4* experiment contain lineages derived from more than one precursor. The number of marked cells per ovariole was roughly consistent with the output from two single-cell lineages in MARCM clones initiated at 0 hr APF. We therefore calculated the component single clone types (frequency and number of products) that would produce the observed *bond-GAL4> G-trace* results if every ovariole had exactly two marked lineages (see Materials and methods). The inferred single precursor lineage types were similar to those induced by *hs-flp* at 0 hr APF (*Figure 11E*, *Table 3*). The most significant differences were a greater yield of ECs per EC-only lineage and fewer FC-only lineages for lineages induced by *bond-GAL4*; both differences are consistent with slightly earlier clone initiation.

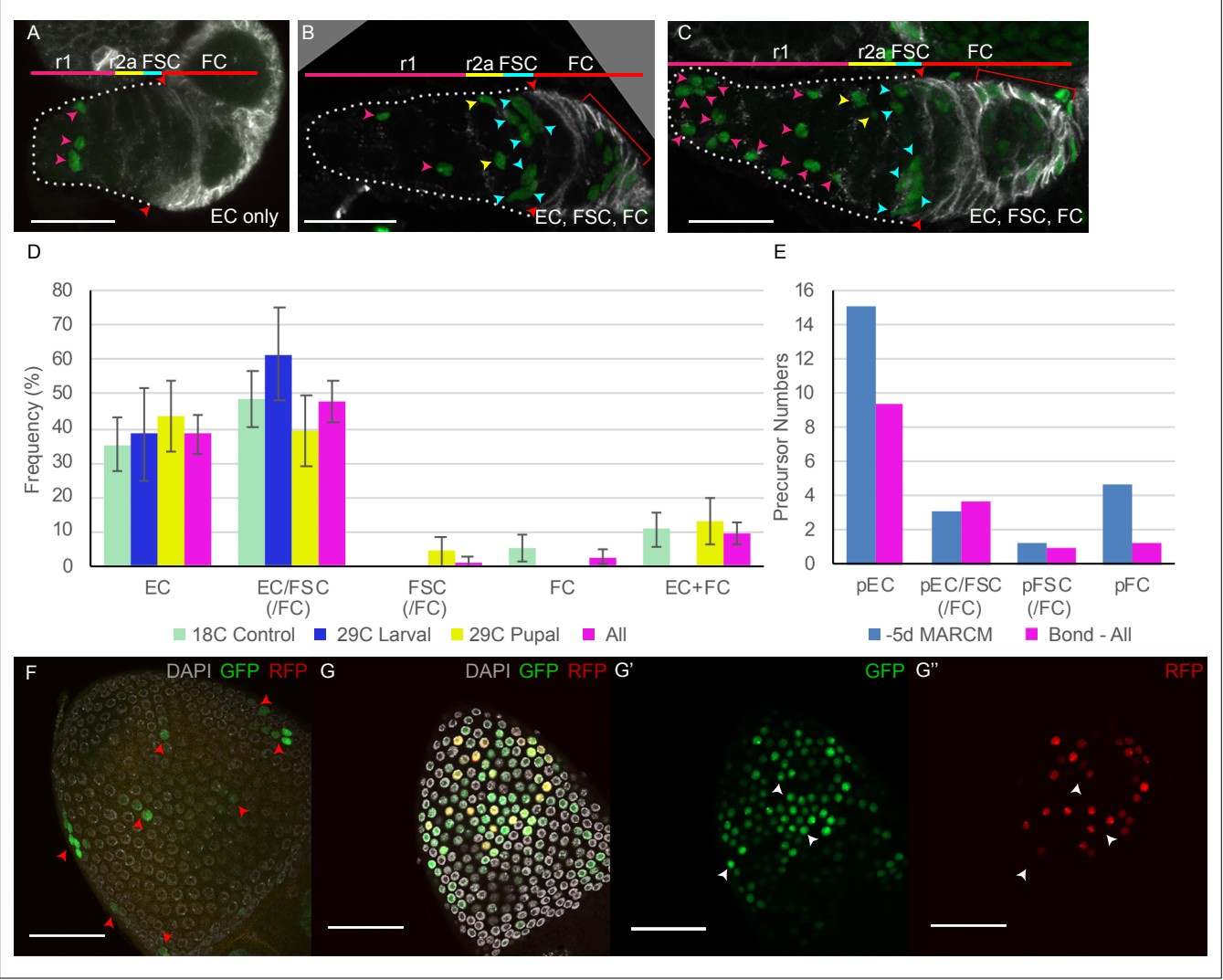

**Figure 11.** Lineages induced by bond-GAL4 reveal typical intermingled cell (IC) precursor patterns and a later extra-germarial crown (EGC) contribution. (**A–G**) *bond-GAL4/G-TRACE* clones (green) were examined in 0–12 hr adult ovaries, stained with antibodies to Fas3 (white in **A–C**) and DAPI (white in **F, G**). (**A–C**) For animals maintained at 18°C throughout development, we observed mainly (**A**) marked escort cells (ECs) only or (**B, C**) marked ECs, follicle stem cells (FSCs), and follicle cells (FCs); scale bars, 20 μm. (**D**) Distribution of clone types observed for animals kept at 18°C throughout (green), raised at 29°C and moved to 18°C 2d after pupariation ('29C larval,' blue), raised at 18°C and moved to 29°C for the last 3–4d of pupation ('29C pupal,' yellow), or the aggregate of all results (pink). EC/FSC and FSC categories almost always included FCs also. (**E**) Estimates of precursor numbers for MARCM clones induced 5d before eclosion (blue) and for the aggregate of all *bond-Gal4* lineages (pink) after disaggregating raw data into single lineages (see Materials and methods). (**F**) Posterior egg chamber (29°C larval) with small GFP clones (arrowheads). Multiple z-sections are projected to show all clones. (**G**) Posterior egg chamber (29°C pupal) with patches of (**G'**) GFP-positive cells, some of which also express (**G"**) RFP, indicating relatively recent expression of *bond-GAL4*. (**F, G**) Scale bars, 50 μm.

The online version of this article includes the following figure supplement(s) for figure 11:

**Source data 1.** Numerical data for graphs in *Figure 11D*.

**Source data 2.** Numerical data for graphs in *Figure 11E*.

It is therefore likely that *bond-GAL4* labeled precursors sporadically throughout the IC region, mostly shortly before pupariation. Consistent with this inference, *bond* RNA and *bond-GAL4* were detected throughout the IC region in mid-third-instar larvae (*Slaidina et al., 2020*) and in some late third-instar larvae (*Banisch et al., 2021*). In keeping with our other lineage studies, inferred single-cell lineages that contained marked FSCs also included marked ECs in almost all cases. The source of lineages reported to include FSCs in the earlier study (*Slaidina et al., 2020*) may also have been pre-pupal ICs (rather than the intended cells posterior to ICs that are selectively labeled by *bond* RNA

just before pupariation) and the ovarioles with both marked ECs and FSCs may have derived from common precursors, consistent with all of our studies, rather than a combination of EC-specific and FSC/FC-specific precursors, as suggested by the authors (*Slaidina et al., 2020*).

When animals with *bond-GAL4* and *G-trace* were maintained at 29C during the last 3d or 4d of pupation, we observed an additional type of lineage, either in isolation or together with the types of marked cells described above (55/88 ovarioles; 0/108 when at 18C throughout) (*Figure 11G*). The second lineage pattern consisted of a few isolated FCs or small FC clusters in the most mature egg chamber, suggesting several independent, recently initiated lineages. Some of the GFP-labeled FCs also expressed *UAS-RFP*, indicating current or recent *bond-GAL4* activity (*Figure 11G*). These patterns were not reported previously (*Slaidina et al., 2020*), presumably because the first-formed egg chambers were no longer present in the 2d-old adults examined. The same type of clone was observed if animals were kept at 29C until about 2d after pupariation (30/44 ovarioles, with only six ovarioles also including RFP-expressing FCs) (*Figure 11F*). The individual patches of marked FCs were much smaller than observed for lineages induced by *hs-flp* at 36 hr APF and therefore presumably result from the initiation of multiple lineages at a later time due to perduring *bond-GAL4* or *UAS-flp* products. The second pattern of lineages initiated by *bond-GAL4* suggests selective expression of *bond* in FCs of the first-formed egg chamber in late pupae and adults. We have no evidence that *bond-GAL4* is expressed sufficiently in precursors of those FCs to initiate lineages earlier in pupal development, potentially reflecting the posterior domain of *bond* RNA expression observed in late third-instar larvae (*Slaidina et al., 2020*).

## Discussion

An important general objective, exemplified here by FSCs, is to ascertain the origin of adult stem cells and surrounding niche cells in order to understand how their number and organization are instructed during development. Our investigations of changing morphology, cell movements, and cell lineage in the *Drosophila* ovary reveal a gradual and flexible adoption of these cell types according to AP location throughout pupal development, rather than early definitive cell fate decisions (*Figure 5*). This contrasts with the development of many other types of differentiated adult cells.

Specifically, adult EC niche cells and FSCs develop from a common group of dividing precursors (ICs) that are interspersed with developing germline cysts. ICs express TJ and are located posterior to cap cells, which serve as adult niche cells for both GSCs and FSCs, and are already specified by the start of pupation. The most anterior ICs become regional subsets of ECs (r1 and r2a), which likely have distinctive roles in the adult (*Shi et al., 2021*; *Tu et al., 2021*). These precursors markedly attenuate division, starting from the most anterior locations, during the second half of pupation (*Table 2*, *Figure 10*, *Figure 1—figure supplement 2*). Thus, the characteristic quiescence of adult ECs is acquired gradually and in a spatially graded manner throughout most of pupation. ICs further from the anterior produce both ECs (mainly r2a) and FSCs (as well as FCs) (*Figure 5*). More posterior ICs produce FSCs and FCs (*Figure 5*), while the most posterior ICs produce only FCs. There is a progressive restriction of fates over time. That likely reflects simply the reduced width of AP territory occupied by derivatives of a single precursor cell as development proceeds, rather than any discrete and irreversible cell specification event. This pattern of germarium development ensures the juxtaposition of ECs and FSCs and may contribute to producing those cell types in appropriate proportions, with EC numbers limited by a progressive decline in proliferation that spreads from the anterior.

Germarium development is likely driven initially by the early establishment of TF and cap cells as a source of anterior organizing signals. The behavior of adult FSCs, including conversion to FCs, additionally relies on a signaling source posterior to the germarium, namely the anterior polar cells of the most recently budded egg chamber (*Figure 1A*; *Melamed and Kalderon, 2020*; *Vied et al., 2012*; *Torres et al., 2003*). Polar cells were observed on the first budded egg chamber by 60 hr APF (*Figure 9—figure supplement 4E and E'*). However, at the start of pupation there are no budded egg chambers, so formation of the first FCs and the first egg chamber must proceed by a different route. We found that those first FCs derive from precursors that have accumulated posterior to ICs by 36 hr APF and differ from ICs by expressing Fas3, by lack of association with germline cells and by being restricted to produce only FCs, and not ECs or FSCs. Thus, precursors of the first FCs, some of which will become polar cells that act as posterior niche cells, have segregated from FSC precursors by 36 hr APF. After both anterior and posterior organizing centers for adult FSCs have been established and

two egg chambers have budded, the architecture and Fas3 expression patterns of the germarium resemble those of adults, suggesting that the developmental process for FSCs and niche cells is complete.

Over the course of pupal development, the entire somatic cellular complement of the adult tissue that is subsequently maintained by stem cell activity derives from an initially small set of precursors that also produce adult FSCs, with outcomes dictated simply by progressive AP cell positioning. Preliminary understanding of other adult stem cells suggests that derivation of tissues and their stem cells from a common source may be frequent but not universal. There is evidence that both adult mouse gut stem cells and adult neural stem cells of the dentate gyrus derive from a precursor pool that also gives rise to the adult epithelial tissue maintained by the stem cell (*Berg et al., 2019*; *Guiu et al., 2019*). By contrast, subventricular zone neural stem cells appear to be specified during embryogenesis and remain quiescent, while the adult tissue it can replenish is built from other precursors (*Fuentealba et al., 2015*; *Furutachi et al., 2015*).

## Defining and locating EC, FSC, and FC precursors through pupal development

This study examined *Drosophila* ovary development comprehensively during pupal stages through lineage analyses initiated at different times, and by systematic analysis of fixed specimens and live imaging. Each experimental approach included some challenges and leaves some issues unresolved, but several key findings appear conclusive and supported by all three approaches.

The challenge of labeling only a single precursor among many is common in lineage studies. Here it was met by using very mild heat-shocks to initiate clones through *hs-flp* induction and a multicolor labeling technique that reduces the frequency of a specific color combination substantially relative to single-color clones. We also estimated the frequency of single-cell lineages from both the overall frequency of ovariole labeling and the frequency of clone patterns (exceptional, discontinuous clones), allowing us to derive mathematical estimations of strictly single-cell lineage frequencies and yields. The results led to robust conclusions that many precursors at the start of pupation yield only ECs, a smaller proportion yield both ECs and FSCs (as well as FCs), and very few produce FSCs with no ECs. Common precursors for ECs and FSCs persist for at least the first 72 hr of pupation (later times were not tested), although they are exceeded in number at 72 hr APF by precursors that produce FSCs but not ECs. Almost all FSC lineages also included marked FCs in newly eclosed adults.

Deducing the locations of precursors required integrating lineage and morphological studies. Each lineage was restricted in its AP extent, but the whole collection of lineages included overlapping domains that spanned the whole length of the ovariole of a newly eclosed adult, from the most anterior r1 EC to FCs and the basal stalk of the terminal egg chamber, suggesting that precursors are arranged in a continuum along the AP axis throughout pupal development (*Figure 5*). We directly observed EdU labeling in the most anterior ICs in early pupae, so we can deduce that the most anterior of the inferred continuum of dividing precursors subject to lineage labeling lie immediately adjacent to cap cells and give rise to adult r1 ECs. The posterior extent of ovariole precursors was deduced by direct observation of the production of the first FCs, as summarized below. Those spatial insights also allowed us to deduce that FSC precursors are ICs, interspersed with germline cells in the germarium throughout pupal development.

## Live imaging reveals the location of precursors of the first FCs

The precursors that give rise to FCs surrounding the most posterior egg chamber of newly eclosed adults were identified by a combination of fixed and live images. These cells all express Fas3 strongly by 24 hr APF and comprise two distinguishable groups. We named one group as extra-germarial crown (EGC) cells in recognition of their location, appearance and TJ expression. Cells of the second, more posterior group do not express TJ and are largely intercalated into a stalk of 1–2 cell's width at 48 hr APF prior to egg chamber budding. During budding, the most mature germline cyst leaves the germarium devoid of accompanying ICs, which do not yet express Fas3. The cyst then enters EGC territory and moves a considerable distance from the germarium as it is enveloped by both EGC and basal stalk cells.

Towards the end of the budding process, cells posterior to the new egg chamber are stacked in single file, forming a basal stalk, which is stably retained into adulthood. We observed several lineages

that included labeled cells in the adult basal stalk and in an FC patch on the terminal egg chamber, reflecting contributions of pupal basal stalk cells. Some EGC derivatives also remain on the first budded egg chamber, while others migrate anteriorly beyond the budded cyst to form a secondary EGC between the germarium and the budded egg chamber. Secondary EGCs provide cells to partially coat the second cyst to emerge from the germarium. The second cyst appears from live imaging to move out of the germarium together with some associated ICs. Lineage studies also suggest that FCs of the second budded egg chamber derive from both EGC cells (producing FCs in egg chambers 4 and 3) and ICs (producing FCs in egg chamber 3 and more anterior locations) (*Figure 6E*).

There is no longer a recognizable EGC after two egg chambers have budded, and the two most mature germline cysts are surrounded by strongly Fas3-positive cells within the germarium, as in adults. Budding of the third, and subsequent, egg chamber is therefore likely similar to the process in adults, with cysts emerging from the germarium completely encased by FCs. Consistent with this inference, the two most posterior germarial cysts are surrounded by Fas3-positive IC cells by the time the second egg chamber starts to bud. Precursors of those FC-forming ICs were almost certainly resident in the germarium throughout pupation because we never observed cells in the expanding EGC population entering the germarium. FSC precursors, which are anterior to FC precursors, must therefore also be ICs, residing within the germarium throughout pupation. This important deduction is supported also by consideration of cell numbers. Over 60 TJ-positive somatic ICs were seen in the germarium as the first germline cyst leaves the germarium at about 48 hr APF (*Figure 7D*). This exceeds the total number of ECs (about 40 on average) plus FSCs (about 16) in an adult germarium, even though many of these precursors are still dividing and amplifying towards producing a full complement of adult ECs and FSCs. Thus, the ICs in the germarium at about 48 hr APF must include all EC and FSC precursors, as well as a significant number of FC precursors.

## Deductions from targeted lineage analyses

A recent study used single-cell RNA sequencing to explore the diversity of precursors in late larval ovaries. It was suggested that a distinct group of cells posterior to ICs and characterized by *bond* expression give rise to FSCs and FCs but not ECs (*Slaidina et al., 2020*). We repeated the supporting studies of lineages initiated from cells expressing *bond-GAL4*. By scoring FSCs directly, counting the number of labeled ECs and FSCs in each sample, and comparing the results to those for heat-shock induced single-cell lineages, we were able to deduce that ovarioles almost always harbored more than one lineage initiated by *bond-GAL4*. This allowed us to estimate the composition of single-cell lineages, and we found that they were very similar to those induced by marking any dividing cell at pupariation. This inference is compatible with the observation that *bond* RNA and *bond-GAL4* are expressed at low levels throughout the IC region in mid-third-instar larvae (*Slaidina et al., 2020*) and with RNA in situs showing *bond* RNA in posterior IC regions of late third-instar larvae (*Banisch et al., 2021*). In those lineages, which appear to be initiated throughout the IC region, rather than specifically in cells posterior to ICs, we found that labeled FSCs almost always included labeled ECs, compatible with a common precursor. It was previously assumed that *bond-GAL4* lineages derived selectively from cells posterior to ICs in late third-instar larvae and that the presence of both labeled ECs and FSCs in the same ovariole resulted from the simultaneous presence of a lineage dedicated to production of each cell type (*Slaidina et al., 2020*). We did find evidence that *bond-GAL4* also targets cells that become FCs of the first-formed egg chamber very late in pupal development. It remains uncertain whether *bond* might be a selective marker of EGC or basal stalk FC precursors during the first half of pupation.

## Coordination between germline and somatic ovarian cell development

Ovary development not only organizes suitable numbers of FSCs, GSCs, and their niche cells into an adult germarium, ready to fuel lifelong adult oogenesis, but also initiates oogenesis early, so that the first eggs can be laid shortly after eclosion. This requires organized progression of germline development. Ecdysone acts positively in somatic cells of late third-instar larvae to initiate GSC differentiation, manifest by induction of the differentiation factor, Bag-of-marbles (Bam) and then cystocyte formation, revealed by the transition of the rounded spectrosome to a branched fusome (*Gancz and Gilboa, 2013*; *Gancz et al., 2011*). Prior studies did not reveal whether Bam-GFP expression or PGC division initiates simultaneously in all PGCs distant from cap cells or in an anterior-posterior gradient,

or how germline differentiation proceeds thereafter. We found that there is a clear anterior to posterior polarity of increasingly developed cysts that produced a mature steady-state pattern by 48 hr APF. This was deduced by counting the number of germline cells linked by fusomes, with 16 cell cysts first appearing 18 hr APF, and by observing the exclusion of EdU DNA replication signals from the most posterior regions of the germarium from 36 hr onwards, indicating occupancy by only 16 cell cysts. Moreover, a single posterior 16-cell cyst emerged for the first time between 36 hr and 48 hr APF.

These observations raise the question of what underlies the spatially organized initiation or progression of differentiation of PGCs that are in different locations at the larval/pupal transition. The BMP signals that maintain anterior pre-GSCs are likely highly spatially restricted, as in adults (*Harris and Ashe, 2011*), while hormonal ecdysone signals may be spatially uniform, suggesting that there must be other important differentiation signals. Potential sources of relevant signals are the adjacent somatic cell precursors, which will later become ECs. There is some evidence that the ordered posterior movement of germline cysts in adult germaria depends on EC interactions (*Banisch et al., 2017*). Perhaps pupal EC precursors, even while still dividing during early pupation, have spatially graded adhesive properties or send graded differentiation signals according to their AP position.

## Organizing patterned development of adult stem cells and niche cells

It is common to think of developmental processes and cell types in terms of establishment of transcriptional networks and key markers that bear witness to permanent decisions. Investigations of adult FSCs, and other adult stem cell paradigms such as the mammalian gut, as well as the observations described here regarding FSC development, reveal a more flexible process, in which environmental signals reflecting current cell locations may be more significant than stable expression of master regulators of transcription (*Reilein et al., 2017*; *Melamed and Kalderon, 2020*; *Beumer and Clevers, 2020*). The transcription factor TJ is selectively expressed in many developing and mature ovarian somatic cells, and it has been shown to be important for the specification of cap cells and many IC derivatives (*Lai et al., 2017*; *Panchal et al., 2017*). We therefore suspected that EGC cells, posterior to ICs, might be FC precursors because they also expressed TJ. Although that suspicion was supported by live imaging studies, the same studies showed that TJ-negative basal stalk cells also became FCs. Thus, TJ neither defines ICs nor EC/FSC/FC precursors. We think that it is likely that studying the spatial distribution of external signals and their interpretation by somatic cells in the developing ovariole will be essential to understand why some cells initially mix with germline cells to become ICs, while others segregate away, and how these cells become ECs, FSCs, and FCs just in time to serve the appropriate functional role.

# Materials and methods

**Key resources table**

| Reagent type (species) or resource | Designation | Source or reference | Identifiers | Additional information |
|---|---|---|---|---|
| Gene (*Drosophila melanogaster*) | sha | FlyBase | FlyBase ID: FBgn0003382 | |
| Gene (*Drosophila melanogaster*) | bond | FlyBase | FlyBase ID: FBgn0260942 | |
| Genetic reagent (*Drosophila melanogaster*) | NM FRT40A | BDSC BL-1835 | RRID:BDSC_1835 | Control for MARCM clones |
| Genetic reagent (*Drosophila melanogaster*) | FRT82B NM | PMID:23079600 | | Control for MARCM clones |
| Genetic reagent (*Drosophila melanogaster*) | FRT42D tub-GAL80 | PMID:28414313 | | For 2R MARCM clones |
| Genetic reagent (*Drosophila melanogaster*) | FRT82B tub-GAL80 | BDSC BL-5135 | RRID:BDSC_5135 | For 3R MARCM clones |
| Genetic reagent (*Drosophila melanogaster*) | *FRT42D act-GAL80 tub-GAL80* | PMID:33135631 | | For 2R MARCM clones at 29C |

*Continued on next page*

*Continued*

| Reagent type (species) or resource | Designation | Source or reference | Identifiers | Additional information |
|---|---|---|---|---|
| Genetic reagent (*Drosophila melanogaster*) | yw *hs-Flp, UAS-nGFP, tub-GAL4* | PMID:33135631 | | For MARCM clones |
| Genetic reagent (*Drosophila melanogaster*) | *ubi-GFP FRT40A* | BDSC BL-5189 | | |
| Genetic reagent (*Drosophila melanogaster*) | *ubi-GFP FRT40A FRT42B His2Av-RFP/tub-lacZ FRT40A FRT42B* | PMID:28414313 | | For multicolor clones |
| Genetic reagent (*Drosophila melanogaster*) | C587-GAL4 | BDSC BL-67747 | RRID:BDSC_67747 | GAL4 expressed in ECs and FSCs |
| Genetic reagent (*Drosophila melanogaster*) | UAS-FUCCI On second chromosome | BDSC BL-55121 | | Nuclear GFP and RFP FUCCI |
| Genetic reagent (*Drosophila melanogaster*) | UAS-FUCCI On third chromosome | BDSC BL-55122 | | Nuclear GFP and RFP FUCCI |
| Genetic reagent (*Drosophila melanogaster*) | *bond-GAL4/G-TRACE* | PMID:31919193 | | Dr. Ruth Lehmann (MIT) |
| Antibody | Anti-GFP (rabbit polyclonal) | Molecular Probes | A6455 | (1:1000) |
| Antibody | Anti-Fas3 (mouse monoclonal) | Developmental Studies Hybridoma Bank (DSHB) | 7G10 | (1:250) |
| Antibody | Anti-Lamin-C (mouse monoclonal) | DSHB | LC28.26 | (1:50) |
| Antibody | Anti-Hts (mouse monoclonal) | DSHB | 1B1 | (1:20) |
| Antibody | Anti-Traffic Jam (guinea pig polyclonal) | PMID:23720044 | RRID:AB_2567862 | Dr. Dorothea Godt (University of Toronto) (1:5000) |
| Antibody | Anti-Zfh-1 (rabbit polyclonal) | Ruth Lehmann | | Dr. Ruth Lehmann (MIT) (1:5000) |
| Antibody | Anti-Castor (rabbit polyclonal) | PMID:9436984 | | Dr. Ward Odenwald (National Institutes of Health) (1:1000) |
| Antibody | Anti-Vasa (rabbit polyclonal) | Ruth Lehmann | | Dr. Ruth Lehmann (MIT) (1:1000) |
| Antibody | Alexa Fluor 488 goat anti-rabbit (polyclonal) | Thermo Fisher Scientific | Cat# A11034; RRID:AB_2576271 | (1:1000) |
| Antibody | Alexa Fluor 647 goat anti-rabbit (polyclonal) | Thermo Fisher Scientific | Cat# A21245; RRID:AB_2535813 | (1:1000) |
| Antibody | Alexa Fluor 594 goat-anti mouse (polyclonal) | Thermo Fisher Scientific | Cat# A11032; RRID:AB_2534091 | (1:1000) |
| Antibody | Alexa Fluor 546 goat-anti mouse (polyclonal) | Thermo Fisher Scientific | Cat# A11030; RRID:AB_2534089 | (1:1000) |
| Antibody | Alexa Fluor 647 goat-anti mouse (polyclonal) | Thermo Fisher Scientific | Cat# 21236; RRID:AB_2535805 | (1:1000) |
| Antibody | Alexa Fluor 647 goat anti-rat (polyclonal) | Thermo Fisher Scientific | Cat# 21247; RRID:AB_141778 | (1:1000) |
| Antibody | Alexa Fluor 488 goat anti-guinea pig (polyclonal) | Thermo Fisher Scientific | Cat# A11073; RRID:AB_2534117 | (1:1000) |
| Antibody | Alexa Fluor 594 goat anti-guinea pig (polyclonal) | Thermo Fisher Scientific | Cat# A11076; RRID:AB_2534120 | (1:1000) |
| Antibody | Alexa Fluor 647 goat anti-guinea pig (polyclonal) | Thermo Fisher Scientific | Cat# A21450; RRID:AB_2735091 | (1:1000) |

*Continued on next page*

*Continued*

| Reagent type (species) or resource | Designation | Source or reference | Identifiers | Additional information |
|---|---|---|---|---|
| Chemical compound, drug | DAPI Fluoromount-G | Southern Biotech | 0100-20 | Mount samples and stain for DAPI |
| Commercial assay or kit | Click-iT EdU Cell Proliferation Kit | Thermo Fisher Scientific | C10086 | |
| Software, algorithm | ZEN Blue, ZEN Black, and ZEN Lite | Zeiss | | For viewing z-stack images and quantifying fluorescence |

## Staging of pupae

Third-instar larvae were sorted to select females, transferred to a vial with food, and checked every hour to mark the time each individual developed into a puparium (white, but immobile, with small anterior spiracles). This time is 0 hr APF. We found that keeping animals in the light at night and in the dark during the day on a 12 hr/12 hr cycle allowed more of them to pupate during the daytime. We used a programmable outlet timer (Nearpow) together with a manual LED soft white nightlight (Energizer Cat# 37099) installed in a dark incubator at 25C .

## Dissection and staining of pupal ovaries

Pupae were removed from the vial wall by adding a drop of water and transferred with forceps into a well of a Corning PYREX glass spot plate (Corning, Cat# 7220-85) containing either PBS or 4% paraformaldehyde solution with care taken not to damage or pierce the posterior end of the pupa. If the dissection was done in fixative solution, it could take no longer than 10 min (ovaries were rocked in clean fixative for a minimum of 15 min, but for no longer than 30 min). Ovaries from 72 hr APF and older had fewer tears in them when dissected directly into fixative. Using a dissecting micro-scope, pupae were held against the bottom of the well using forceps and the posterior tip was cut off using Vannas Spring Scissors (2.5 mm straight edge, Fine Science Tools, Foster City, CA, item number 15000-08) about a third of the way in from the posterior end. The anterior part of the pupa was removed to a discard well. The posterior end was searched for ovaries (clear, spherical, striated structures) by gently dislodging fat tissue surrounding the ovaries so as not to cloud the well with debris. This was done by gently tearing apart the contents of the pupal case or swirling them in the well using two pairs of forceps. As soon as they were spotted, ovaries were transferred to a new well of fixative by the following process. A 20 µl pipette tip with a pipettor set to 20 µl was coated in 10% Normal Goat Serum (NGS) by pipetting up and down several times. The pipettor was set to 5 µl to pipette up the ovary from the dissection well to transfer into the fixative well. Ovaries were fixed by rocking in the well for 15 min at room temperature. Fixative was removed with a 1000 µl pipette set to 270 µl to slowly and carefully pipette up the liquid. The 1000 µl pipettor was used to remove liquid after all subsequent steps. Ovaries were rinsed 3× in 2%  PBST 0.5%  Tween solution for 5, 10, and 45 min at 4C  by rocking very slowly in a horizontal plane. The glass dissection dish was covered with a large pipette tip box. Ovaries were rinsed for 30–60 min in 10%  NGS solution at 4C , and then rocked gently in primary antibody overnight. Ovaries were rinsed 3×  with 0.5%  PBST and then incubated for 1 hr (covered, cold room, rocker set on 2) in secondary antibody diluted 1:1000 with 0.5%  Triton. Ovaries were rinsed 2× with 0.5%  PBST and 1× with PBS. To mount ovaries, a 20 µl pipette tip coated in 10%  NGS solution as described above was set to 5 µl and used to capture the ovary and transfer to a glass slide. 20 µl of DAPI Fluoromount was added to a coverslip, and then placed on top of the slide with the ovaries. Care was taken to avoid pressing the coverslip and potentially disfiguring the mounted ovaries. For older pupal ovaries, specifically 72 hr APF and onward, forceps were used to gently tear apart the ovarioles to separate germaria for clearer imaging. A sharpie was used to draw arrows to mark the location of the ovaries on the coverslip.

## EdU labeling

Ovaries were dissected directly into 15 µM EdU in Schneider's insect medium, incubated for 1 hr at room temperature, and then fixed for 10 min at room temperature in 4%  paraformaldehyde. EdU incorporation was detected using the Click-iT Plus EdU Imaging Kit C10637 (Life Technologies).

## Immunohistochemistry

Monoclonal antibodies against Lamin-C, Fasciclin III, Vasa, and Hts were obtained from the Developmental Studies Hybridoma Bank, created by the NICHD of the NIH and maintained at The University of Iowa, Department of Biology, Iowa City, IA. 7G10 anti-Fasciclin III was deposited to the DSHB by C. Goodman and was used at 1:300 for multicolor lineage experiments and at 1:250 in all other stainings. 1B1 anti-Hts was deposited by H.D. Lipshitz. LC28.26 anti-Lamin-C at 1:50 was deposited by P.A. Fisher. Anti-Vasa (used at 1:10) was deposited by A. C. Spradling/D. Williams and used for staining adult ovaries. For pupal ovaries, the monoclonal rat anti-Vasa was not effective and instead rabbit anti-Vasa (gift from R. Lehmann, NYU Medical School) was used. Other primary antibodies used were rabbit anti-Zfh-1 at 1:5000 (a gift from R. Lehmann, NYU Medical School), guinea pig anti-Traffic Jam at 1:5000 (a gift from Dorothea Godt, University of Toronto, Canada), anti-β-galactosidase (Cat# 55976, MP Biomedicals) at 1:1000; anti-GFP (A6455, Molecular Probes) at 1:1000; goat FITC-anti-GFP (Abcam ab6662) at 1:400. Secondary antibodies were Alexa-488, Alexa-546, Alexa-594, or Alexa-647 from Molecular Probes. Ovarioles from multicolor experiments were mounted in Prolong Gold Antifade (Invitrogen). DAPI-Fluoromount-G (Southern Biotech) was used as mounting medium for all other experiments. Images were collected with a Zeiss LSM700 or LSM800 laser scanning confocal microscope (Zeiss) using a 63 × 1.4 N.A. lens.

## Adult ovary fixation and staining

Ovaries were fixed in 4% paraformaldehyde in PBS for 10 min at room temperature, rinsed three times in PBS with 0.1% Triton and 0.05% Tween-20 (PBST), and blocked in 10% normal goat serum (Jackson ImmunoResearch Laboratories) in PBST. Ovarioles were incubated in primary antibody for 45 min for multicolor lineage experiments and overnight for all other experiments. Ovarioles were rinsed in PBST three times and incubated for 1 hr in secondary antibodies diluted 1:1000.

## Live imaging and analysis

Live imaging was performed as in *Reilein et al., 2018a*, except that pupal ovaries dissected in imaging medium were left intact instead of separating into individual ovarioles. Intact pupal ovaries were mixed with Matrigel in a well of a gas-permeable imaging chamber. Ovaries were of genotype *yw hs-flp; ubi-GFP FRT40A* (Ubi-GFP ovaries) or *yw hs-flp/yw; ubi-GFP FRT40A FRT42B His2Av-RFP/ tub-lacZ FRT40A FRT42B* (referred to as multicolor ovaries). Clones were generated in the multicolor ovaries by two 10 min heat-shocks at 37C spaced 8 hr apart, 48–72 hr before imaging in order to generate GFP-only and RFP-only cells that were easier to track. Images were acquired every 15 min with 2.5 μm between z-stacks. Volocity Software (Quorum Technologies, Puslinch, ON, Canada) was used to make the 4D reconstruction of the Ubi-GFP labeled ovary. All nineteen z slices covering 45 μm were used in the reconstruction; however, germaria drifted during imaging such that FCs covering approximately one quarter of the budded egg chamber on the lower germarium were not imaged. Cells were colored with Procreate software (Savage Interactive, North Hobart, Australia). Germline cells in the Ubi-GFP movie were identified based on their larger size and paler labeling with GFP compared to somatic cells.

## Clonal analysis and FUCCI labeling genotypes

MARCM clones for lineage analysis were generated on 2R and 3R using the following genotypes:

> (2R ) yw *hs-Flp, UAS-nGFP, tub-GAL4/ yw; FRT42D act-GAL80 tub-GAL80/ FRT42D sha; act> CD2> GAL4/+* and (3R ) yw *hs-Flp, UAS-nGFP, tub-GAL4/ yw; act> CD2> GAL4 UAS-GFP/+; FRT82B tub-GAL80/FRT82B NM*.
> For multicolor clones, flies were of the genotype: *yw hs-flp/yw; ubi-GFP FRT40A FRT42B His2Av-RFP/tub-lacZ FRT40A FRT42B*.
> Genotypes for FLY-FUCCI were *C587-GAL4/ yw; UAS-FUCCI/Cyo; UAS-FUCCI/ (UAS-FUCCI or TM6B)*.

## Lineage analysis by MARCM

Pupae were heat-shocked at 33C for 12 min at the number of days indicated before eclosion. Flies were dissected on the day of eclosion (0d MARCM) or collected on the day of eclosion and dissected

2 days later. Ovaries were stained using antibodies to Vasa, Fas3, and GFP. Cells were scored as r1 or r2a based on measurements that were performed to determine the lengths of these regions. To determine the boundaries of each region, we examined EdU incorporation in ovaries of newly eclosed flies and measured the distance from cap cells to the end of region 1 as determined by the presence of cysts that incorporated EdU. Region 1 cysts are going through mitosis and region 2a cysts have the complete complement of 16 germ cells. The length ratio of region 1:2a is about 3:1 in newly eclosed adults. To measure the percentage of each egg chamber covered by marked FCs, GFP was used to count the number of marked FCs and DAPI was used to count the total number of FCs in each z-section.

### *bond-GAL4/G TRACE* experiments

*bond-GAL4/G-TRACE* flies (a gift of Ruth Lehmann, NYU medical center) were crossed at 18C and either kept at 18C , shifted after egg-laying to 29C and shifted back to 18C in mid-pupation (flies took six more days to eclose at 18C ); or shifted to 29C 3–4 days before eclosion. Flies of geno-type *UAS-RFP, UAS-flp, ubi>stop> GFP/+; bond-GAL4/ tub-GAL80(ts)* were dissected on the day of eclosion.

### Inferring clone types, FSC and EC numbers in newly eclosed adults from scoring 2d-old adults

In order to make quantitative conclusions about the developmental process up to eclosion from scoring clonal outcomes in 2d-old adults, we made the following assumptions and adjustments. We assumed that EC numbers and locations did not change during this 2d interval because ECs generally appear to be quite long-lived. ECs are, however, produced from FSCs during adulthood, so it is likely that a small number of marked ECs scored at +2d were produced from FSCs after eclosion. The proportion of marked ECs arising from FSCs is likely small based on known rates of FSC conversion to ECs (*Melamed and Kalderon, 2020*; *Reilein et al., 2018b*) and the observed high frequency of FSC clones with no ECs (44%) from –2d samples. Individual marked FSCs can be lost, mainly by becoming FCs, or amplified at high rates during adulthood (*Reilein et al., 2017*). If one or more marked FSCs were present at 0d but became an FC by +2d , the resulting marked FCs would reside around the two germarial cysts or in the two youngest egg chambers because egg chambers bud roughly every 12 hr, allowing four cycles of FC recruitment over 2d. We therefore scored an ovariole as containing an FSC (classifying it as an FSC-only or FSC/EC lineage, according to EC content) if there were any marked FCs up to and including the second egg chamber, even if there were no FSCs at +2d . In those cases, we scored the number of marked FSCs as zero. On average, the number of marked FSCs should remain constant from 0d to +2d , so we should obtain a very good estimation of FSC numbers at 0d by scoring the numbers of marked FSCs at +2d , provided we include all examples where all marked FSCs were lost as containing zero FSCs. By using these guidelines, we could surmise the number of different types of clone at 0d and the number of constituent ECs and FSCs from data collected by scoring clones in 2d-old adults.

### Calculation of single-cell lineage frequencies assuming independent recombination events

The proportion of marked ovarioles with a lineage originating from a single marked cell is generally estimated by assuming that the probability of each dividing precursor to be labeled is independent. The estimate derives principally from the observed frequency of unlabeled ovarioles, but it is also dependent to a small degree on the number of potential target cells. Different numbers of assumed target cells are used in the calculations below to demonstrate that estimated single-cell frequencies are not greatly altered by this value.

For the –6d to –2d time course, the average frequency of labeled ovarioles was about 60% .

If there were, for example, only six target cells and we assume a binomial distribution (independent probabilities) of labeling, the chance of any one cell giving a clone is p and the chance of six cells not giving a clone is $(1p)^6$ . The observed no-clone fraction was 0.4, so $(1p)$ = 0.86 and $P$ = 0.14. Hence, the single-clone frequency among six cells is $6p(1p)^5$ , and the frequency among labeled ovarioles is 1/0.6-fold higher (because 60% were labeled). Therefore, the expected frequency of single-cell lineages is $6 \times 0.14 \times (0.86)^5/0.6 = 0.66$.

If there were instead 15 target cells, then $P$ = 0.06 (because $0.94^{15}$ = 0.4) and the frequency of single-cell lineages is $15 \times 0.06 \times (0.94)^{14}/0.6$ = 0.63.

If there were 30 target cells, then $P$ = 0.03 (because $0.97^{30}$ = 0.4) and the frequency of single-cell lineages is $30 \times 0.03 \times (0.97)^{29}/0.6$ = 0.62.

For the multicolor experiment examining newly eclosed adults 5d after heat-shock, the frequency of labeled ovarioles was 26/115 = 23% . If there were 20 target cells (likely in the range of the actual number of precursors), then $P$ = 0.013 (because $0.987^{20}$ = 0.77) and the frequency of single-cell lineages is $20 \times 0.013 \times (0.987)^{19}/0.23$ = 0.86.

For the MARCM experiment examining newly eclosed adults 5d after heat-shock, the frequency of labeled ovarioles was 98/207 = 47% . If there were 20 target cells, then $P$ = 0.031 (because $0.969^{20}$ = 0.53) and the frequency of single-cell lineages is $20 \times 0.031 \times (0.969)^{19}/0.47$ = 0.72.

## Calculation of single-cell lineage frequencies assuming frequencies of single lineages based on the experimental frequency of ovarioles with marked ECs and FCs

In MARCM and multicolor lineage experiments scored in newly eclosed adults, we observed a higher frequency of ovarioles with labeled ECs and FCs but no FSCs (which we assume represent two distinct lineages because the marked cells are discontinuous) than would be expected based on the frequency of unmarked ovarioles and assuming all recombination events are independent. Based on the EC/FC clone frequency, we used trial and error (using increments of 5%) to find the percentage of ovarioles with a single lineage that best fit the entire dataset of clone-type frequencies we observed using the simplifying assumption that all other marked ovarioles harbored exactly two lineages. In reality, the proportion of ovarioles with single lineages may be slightly different and a few ovarioles may include three or more marked lineages but the simplifying approximations we used are likely to be close to reality. These calculations, presented below, are better approximations than are generally made by simply considering the proportion of ovarioles with no marked cells.

### MARCM clones induced 5d (0 hr APF) before eclosion
Single-cell lineage frequencies and labeled cell content were calculated after estimating that 30% of marked ovarioles derived from a single marked cell, while the remaining 70% derived from two marked cells. In all cases, EC/FSC and FSC-only categories include clones with and without FCs (aggregated).

pEC, pEC/FSC, pFSC, and pFC are the proportions of each type of single-cell lineage amongst all marked lineages. Observed values are the proportion of marked ovarioles with the specific type(s) of cell. EC-only means no FSC. FC-only means no FSC. Both categories include EC + FC ovarioles.

Observed EC-only = 0.47 + 0.16 (with FCs) = 0.63 = 0.3 pEC + 0.7 (pEC)$^2$ + 1.4 pEC.pFC.
Observed FC-only = 0.09 + 0.16 = 0.25 = 0.3 pFC + 0.7 (pFC)$^2$ + 1.4 pEC.pFC.
Above is satisfied by pEC = 0.63 and pFC = 0.19.
Observed FSC-only = 0.03 = 0.3 pFSC + 0.7 [(pFSC)$^2$ + 2pFSC.pFC] = 0.7[(pFSC)$^2$ + 0.566pFSC].
So, pFSC = 0.050.
Hence, pEC/FSC = 1 – (pEC+ pFSC + pFC) = 0.13.

The above values are the inferred single-lineage frequencies of different types. The number of marked ECs and FSCs in each category is then deduced from these proportions and the observed number of marked cells in different types of marked ovariole.

Observed ECs per EC-only ovariole = (2.4 × 0.47 + 2.9 × 0.16)/ 0.63 = 2.53.
Of double clones in EC-only, 0.397/0.636 = 0.624 are deduced from clone-type proportions to be two EC clones and 0.239/0.636 = 0.376 are one EC and one FC clone.
So, number of EC lineages per EC-only ovariole = 0.3 + 0.7 (1.248 + 0.376) = 1.44.
So ECs per EC-only lineage = 2.53/1.44 = 1.76.
Observed FSCs per FSC-only ovariole = 14/3.
Of double clones in FSC-only, 0.0025/0.215 = 0.116 are two FSC clones and 0.019/0.215 = 0.884 are one FSC and one FC clone.
So, number of FSC lineages per FSC-only ovariole = 0.3 + 0.7 (0.232 + 0.884) = 1.08.
So FSCs per FSC-only lineage = 14/(3 × 1.08) = 4.32.
Total ovarioles with marked cells = 98, so total number of lineages = 98 × 1.7 = 166.6.
Total ECs from EC-only lineages = 166.6 pEC(1.76) = 184.7.

Total FSCs from FSC-only lineages = 166.6 pFSC(4.32) = 35.9.
Total ECs scored = 253, total FSCs scored = 111.
So, total ECs from EC/FSC clones = 68.3.
Total FSCs from EC/FSC clones = 75.1
Total EC/FSC lineages = 166.6pEC/FSC = 21.66.
So, ECs per EC/FSC clone = 3.15, and FSCs per EC/FSC clone = 3.47.
Deduced 24.0 precursors (to produce 16 FSCs): 15.1 pECs, 3.1 pEC/FSCs, 1.2 pFSCs, 4.6 pFCs.

## Multicolor clones induced 5d (0 hr APF) before eclosion

Analogous calculations for multicolor GFP-only clones where EC plus FC category was 8% and we consequently assume that 70% of marked ovarioles have lineages derived from one cell, and 30% have two lineages.

Observed EC-only = 0.423 + 0.077 = 0.50 = 0.7 pEC + 0.3 (pEC)$^2$ + 0.6 pEC.pFC.
Observed FC-only = 0.077 + 0.077 = 0.154 = 0.7 pFC + 0.3 (pFC)$^2$ + 0.6 pEC.pFC.
Above satisfied by pEC = 0.52 and pFC = 0.14.
Observed FSC-only = 0.077 = 0.7 pFSC + 0.3 [(pFSC)$^2$ + 2 pFSC.pFC].
= 0.3[(pFSC)$^2$ + 0.778 pFSC].
So, pFSC = 0.095.
Hence, pEC/FSC = 1 – (pEC + pFSC + pFC) = 0.245.

The above values are the inferred single-lineage frequencies of different types. The number of marked ECs and FSCs in each category is then deduced from these proportions and the observed number of marked cells in different types of marked ovariole.

Observed ECs per EC-only ovariole = 38/13 = 2.92.
Of double clones in EC-only, 0.2704/0.416 = 0.65 are two EC clones and 0.1456/0.416 = 0.35 are one EC and one FC clone.
So, number of EC lineages per EC-only ovariole = 0.7 + 0.3 (1.3 + 0.35) = 1.195.
So ECs per EC-only lineage = 2.92/1.195 = 2.44.
Observed FSCs per FSC-only ovariole = 4.0.
Of double clones in FSC-only, 0.00593/0.02363 = 0.25 are two FSC clones and 0.0177/0.02363 = 0.75 are one FSC and one FC clone.
So, number of lineages per FSC-only ovariole = 0.7 + 0.3 (0.50 + 0.75) = 1.0375.
So FSCs per FSC-only lineage = 4.0/1.0375 = 3.86.
Total ovarioles with marked cells = 26, so total number of lineages = 26 × 1.3 = 33.8.
Total ECs from EC-only lineages = 33.8 pEC(2.44) = 42.89.
Total FSCs from FSC-only lineages = 33.8 pFSC(3.86) = 10.04.
Total ECs scored = 66, total FSCs scored = 29.
So, total ECs from EC/FSC clones = 23.11.
Total FSCs from EC/FSC clones = 18.96.
Total EC/FSC lineages = 33.8 pEC/FSC = 8.28.
So, ECs per EC/FSC clone = 2.79, and FSCs per EC/FSC clone = 2.27.

Deduced 17.3 precursors (to produce 16 FSCs): 9.0 pECs, 4.2 pEC/FSCs, 1.6 pFSCs, 2.0 pFCs.

## MARCM clones induced 3.5d (36 hr APF) before eclosion

Analogous calculations for MARCM clones induced at –3.5d, where EC plus FC category was 44% and assuming all marked ovarioles have two lineages.

Observed EC-only = 0.207 + 0.438 = 0.645 = (pEC)$^2$ + 2 pEC.pFC.
Observed FC-only = 0.091 + 0.438 = 0.529 = (pFC)$^2$ + 2 pEC.pFC.
Above satisfied by pEC = 0.51 and pFC = 0.38.
Observed FSC-only = 0.033 = (pFSC)$^2$ + 2 pFSC.pFC = (pFSC)$^2$ + 0.76 pFSC.
So, pFSC = 0.041.
Hence, pEC/FSC = 1 – (pEC + pFSC + pFC) = 0.07.

The above values are the inferred single-lineage frequencies of different types. The number of marked ECs and FSCs in each category is then deduced from these proportions and the observed number of marked cells in different types of marked ovariole.

Of double clones in FC-only, 0.144/0.532 = 0.27 are two FC clones and 0.388/0.532 = 0.73 are one EC and one FC clone.
Observed ECs per EC-only ovariole = 199/78 = 2.55.
Of double clones in EC-only, 0.260/0.646 = 0.40 are two EC clones and 0.388/0.646 = 0.60 are one EC and one FC clone.
So, number of EC lineages per EC-only ovariole = 0.8 + 0.6 = 1.40.
So, ECs per EC-only lineage = 2.55/1.40 = 1.82.
Observed FSCs per FSC-only ovariole = 13/4 = 3.25.
Of double clones in FSC-only, 0.00168/0.03284 = 0.05 are two FSC clones and 0.03116/0.03284 = 0.95 are one FSC and one FC clone.
So, number of FSC-only lineages per FSC-only ovariole = 1.05.
So, FSCs per FSC-only lineage = 3.25/1.05 = 3.10.
Total ovarioles with marked cells = 121, so total number of lineages = 242.
Total ECs from EC-only lineages = 242 pEC(1.82) = 224.6.
Total FSCs from FC-only lineages = 242 pFSC(3.10) = 30.8.
Total ECs scored = 276, total FSCs scored = 111.
So, total ECs from EC/FSC clones = 51.4
Total FSCs from EC/FSC clones = 80.2.
Total EC/FSC lineages = 242 pEC/FSC = 16.94.
So ECs per EC/FSC clone = 3.03, and FSCs per EC/FSC clone = 4.73.

Deduced 34.9 precursors (to produce 16 FSCs): 17.8 pECs, 2.4 pEC/FSCs, 1.4 pFSCs, 13.3 pFCs. Also, by this time there are EC precursors that no longer divide (estimated at 5 for –4d and 18 for –3d from MARCM lineages examined in +2d adults, so perhaps 11.5 at –3.5d, to give a total of 45.4 precursors). So, estimated proportion of precursors that are pFCs is 13.3/45.4 = 29% .

## *bond-GAL4/G-TRACE* lineages

Single-lineage frequencies and content for clones initiated by *bond-GAL4* were calculated in a slightly different way (considering ovarioles with only marked ECs to calculate EC-only single lineage and then ovarioles with labeled ECs and FCs clones to calculate the FC-only lineage frequency) because there were almost no ovarioles with only marked FCs.

Here we assumed that every labeled ovariole had exactly two lineages.

Observed EC frequency = 38.36% = $(pEC)^2$.
So, pEC = 61.9%.
Observed EC + FC frequency = 9.59% = 2 pEC.pFC.
So, pFC = 7.75%.
Observed FSC frequency = 1.37% = $(pFC)^2$ + 2 pFC. pFSC.
So, pFSC = 6.29%.
Therefore, pEC/FSC = 24.0%.
Observed EC per EC-only (or EC + FC) lineage = 207/35 = 5.91.
Of double clones in EC-only, expected frequencies: 0.383/0.479 = 0.80 are two EC clones and 0.096/0.479 = 0.20 are one EC and one FC clone.
So, number of EC lineages per EC-only ovariole = 2 × 0.8 + 0.2 = 1.8.
So, ECs per EC-only lineage = 5.91/1.8 = 3.28.
Ovariole with labeled FSCs but no ECs has seven FSCs.
Of double clones in FSC-only, 0.00396/0.01371 = 0.289 are two FSC clones and 0.00975/0.01371 = 0.721 are one FSC and one FC clone.
So, number of FSC lineages per FSC-only ovariole = 2 × 0.289 + 0.721 = 1.32.
So, FSCs per FSC-only lineage = 7/1.32 = 5.3.
Total ovarioles with marked cells = 73, so total number of lineages = 146.
Total ECs from EC-only lineages = 146 pEC(3.28) = 296.4.
Total FSCs from FSC-only lineages = 146 pFSC(5.3) = 48.7.
Total ECs scored = 496, total FSCs scored = 190.
So, total ECs from EC/FSC clones = 199.6.
Total FSCs from EC/FSC clones = 141.3.
Total EC/FSC lineages = 146 pEC/FSC = 35.4.
So, ECs per EC/FSC clone = 5.64, and FSCs per EC/FSC clone = 3.99.

## Acknowledgements

This work was supported by NIH RO1 GM079351 to DK. We thank Ruth Lehmann, Dorothea Godt, and Ward Odenwald for reagents, Pegah Khosravi-Kamrani, Shay Mallick, Aaron Choi, and Jack Misner for help with experiments and analysis, David Melamed for continued discussions and input, the Bloomington Stock Center for provision of genetic reagents, the Developmental Studies Hybridoma Bank (DSHB) for antibodies, FlyBase as an information resource, and the confocal microscope resource provided by the Department of Biological Sciences, Columbia University.

## Additional information

### Funding

| Funder | Grant reference number | Author |
| --- | --- | --- |
| National Institutes of Health | GM079351 | Daniel Kalderon |

The funders had no role in study design, data collection and interpretation, or the decision to submit the work for publication.

### Author contributions

Amy Reilein, Conceptualization, Data curation, Formal analysis, Investigation, Methodology, Project administration, Supervision, Validation, Visualization, Writing - original draft, Writing – review and editing; Helen V Kogan, Conceptualization, Data curation, Formal analysis, Investigation, Methodology, Visualization, Writing – review and editing; Rachel Misner, Data curation, Formal analysis, Investigation, Methodology, Validation, Visualization, Writing – review and editing; Karen Sophia Park, Formal analysis, Investigation, Visualization; Daniel Kalderon, Conceptualization, Data curation, Formal analysis, Funding acquisition, Investigation, Methodology, Project administration, Resources, Supervision, Validation, Writing - original draft, Writing – review and editing

### Author ORCIDs

Amy Reilein (ID) http://orcid.org/0000-0001-9464-0102
Helen V Kogan (ID) http://orcid.org/0000-0002-6149-6674
Rachel Misner (ID) http://orcid.org/0000-0001-5814-9359
Karen Sophia Park (ID) http://orcid.org/0000-0002-1605-3110
Daniel Kalderon (ID) http://orcid.org/0000-0002-2149-0673

### Decision letter and Author response

Decision letter https://doi.org/10.7554/eLife.69749.sa1
Author response https://doi.org/10.7554/eLife.69749.sa2

## Additional files

### Supplementary files

• Transparent reporting form

### Data availability

All data generated or analysed during this study are included in the manuscript and supporting files. Source data files are provided for all graphical data.

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
