## [Decision Letter]

**Acceptance summary:**

This manuscript presents a very extensive set of experiments describing the origin and fate of various cell populations in the *Drosophila* ovary. This is a well established and genetically amenable system to explore interactions between adult stem cells and their niches. The paper presents a new view of how different cell types acquire their fates during development. This is a more nuanced view than extant models, involving common progenitors from which different cell fates (stem cell, progeny and niche cells) arise gradually and relying on spatiotemporal cues. It will be an important study for the stem cell/developmental biology field.

**Decision letter after peer review:**

Thank you for submitting your article "Adult Stem Cells and Niche Cells segregate gradually from common precursors that build the adult *Drosophila* ovary during pupal development" for consideration by *eLife*. Your article has been reviewed by 2 peer reviewers, and the evaluation has been overseen by a Reviewing Editor and Anna Akhmanova as the Senior Editor. The reviewers have opted to remain anonymous. We are sorry that it took so long to make this decision: A third reviewer promised their review several times but never delivered it.

Essential revisions:

The reviewers found the work to be convincing although the sheer amount of data tends to hide the significance of the conclusions.

Therefore, the main recommendation for a revised version is that you shorten very significantly the manuscript by moving more data to the supplement, and by explaining the main findings in the figures, and their importance in lay terms in the text. The abstract should also be re-written to better convey the main findings to a wider audience. This would make the paper much more powerful to convey the message.

The reviewers have also a number of smaller questions that you will want to address but will not require significant experiments.

*Reviewer #1 (Recommendations for the authors):*

1. The presentation is too lengthy and some of the figures do not follow the order in the text. For example, Figure 3-5 could be combined for easy comparison, and the detailed images could go to supplementary.

2. The title should indicate the second part of the paper (EGCs and basal stalks) as well. The word "segregate gradually" is not explained well.

3. Summary, "set up a key posterior signaling center for regulating adult FSC behavior", please explain what kind of signaling center?

4. Figure 1, please move 1J upfront.

5. Figure 2, could move to supplementary. Also, the quality of the image does not provide enough information to distinguish the different germline stages. A zoomed in view of the four stages shown by different arrows should be provided.

6. Figure 3B, should be moved to Figure 4, or combine Figure 3-5.

7. Figure 4, showing quantification could be more helpful than showing examples.

*Reviewer #2 (Recommendations for the authors):*

As the paper is very long and dense, we struggled to review it to the standards we normally apply to other manuscripts. I would recommend that the authors shorten the manuscript by moving more data to the supplement, and try to explain the data in the main figures, their findings and importance in lay terms. The abstract could also be re-written to attempt to convey the main findings to a wider audience.

Aside from the general points described in the public review regarding the manuscript's presentation and the suggested genetic interrogation for model validation, we have the following suggestions:

Figure panel labelling. We found the cartoons really helpful but we still got a bit lost with some of the figure panels and had to go back and forth between figures. It would be helpful to, for example, see the labels for specific markers on the figure panels themselves rather than having to check in each figure legend, and insert additional cartoons to orient the readers who might be unfamiliar with the system/cell types/markers.

The authors only discussed the single cell RNAseq paper from the Lehmann lab in the context of bond-Gal4. Can anything else be learned by considering it more "holistically" in the context of the revised lineage relationships revealed by the authors? (e.g. regarding the separate precursors of FCs in different egg chambers, or the gradual restrictions of precursor fates according to cell locations along the AP axis…). Whilst the initial clustering may not have been revealing of these features, further (sub)clustering informed by this new knowledge may reveal some interesting candidate genes/cell subtypes.

---

## [Author Response]

Essential revisions:The reviewers found the work to be convincing although the sheer amount of data tends to hide the significance of the conclusions.Therefore, the main recommendation for a revised version is that you shorten very significantly the manuscript by moving more data to the supplement, and by explaining the main findings in the figures, and their importance in lay terms in the text. The abstract should also be re-written to better convey the main findings to a wider audience. This would make the paper much more powerful to convey the message.The reviewers have also a number of smaller questions that you will want to address but will not require significant experiments.

We thank the Editors and Reviewers for the considerable effort and time invested in appraising our lengthy manuscript and for making suggestions for improvements. The required and suggested changes are detailed under comments from each reviewer. They include moving two and a half Figures to supplementary data, ensuring that each section of text included a clear final summary sentence of the main finding in plain language, a completely revised abstract, ands a large number of text edits (principally deletions) to remove duplications or unnecessary material. There was a limit to such changes, enforced by the need to display the requisite evidence to make rigorous conclusions.

Reviewer #1 (Recommendations for the authors):1. The presentation is too lengthy and some of the figures do not follow the order in the text. For example, Figure 3-5 could be combined for easy comparison, and the detailed images could go to supplementary.

I believe Figures are all in order now. Figures 3-5 (now 2-4) all have specific virtues as complementary approaches (summarized in Figure 2A), where examples of images are key to conveying the messages of AP clustering and lineages spanning borders between cell types. These are key points, requiring direct visibility as main Figures and cannot be compressed into just one or two Figures. We have made accommodations elsewhere instead (moving Figures 2, 11 and half of the former Figure 8 to Supplementary).

2. The title should indicate the second part of the paper (EGCs and basal stalks) as well. The word "segregate gradually" is not explained well.

We appreciate the suggestion but find we cannot convey the second part of the paper within a limited number of words or in a general form. Likewise, it is hard to explain the concept of “segregate gradually” just within a title. We believe that the revised Summary (and Introduction) accomplish that.

3. Summary, "set up a key posterior signaling center for regulating adult FSC behavior", please explain what kind of signaling center?

We have re-written the Summary to explain this.

4. Figure 1, please move 1J upfront.

Thank you- done.

5. Figure 2, could move to supplementary. Also, the quality of the image does not provide enough information to distinguish the different germline stages. A zoomed in view of the four stages shown by different arrows should be provided.

We have done that (now Figure1—figure supplement 1) and key fusome details are mostly directly visible; others required looking at additional z-sections but this has been done to support the stated observations.

6. Figure 3B, should be moved to Figure 4, or combine Figure 3-5.

We moved this (and changed it slightly to illustrate key points in the text better) to become part of the next Figure (now 3A).

7. Figure 4, showing quantification could be more helpful than showing examples.

The quantitation is in the next Figure (now Figure 4D, E), where it is presented together with analogous quantitation of MARCM 0h and 36h APF experiments to allow direct comparisons. We believe the direct image evidence is important and attractive, with offset clones of different color (Figure 3E-G) hard to present in any other way.

Reviewer #2 (Recommendations for the authors):As the paper is very long and dense, we struggled to review it to the standards we normally apply to other manuscripts. I would recommend that the authors shorten the manuscript by moving more data to the supplement, and try to explain the data in the main figures, their findings and importance in lay terms. The abstract could also be re-written to attempt to convey the main findings to a wider audience.

We re-wrote the Summary to convey the most central general point within the word limit, with further elaboration in the Introduction.

We moved two Figures (formerly 2 and 11) to Supplementary plus the second part of current Figure 7 (formerly Figure 8).

We have deleted sections, sentences and phrases throughout.

We have made sure there are suitable interim summaries of main points, charting the general course of main Figures retained. These can be found, in almost all cases, as the last sentence in a section and are in plain language.

The story told in the retained main Figures is outlined below.

Figure 1: general outline of pupal ovary development prior to detailed study.

Figures 2-4; Three lineage approaches to establish common precursors of ECs, FSCs and FCs, together with AP clustering of lineages from the most anterior ECs to the most posterior FCs, as summarized in the cartoon of Figure 5.

Figures 6-9: Lineage studies, fixed and live imaging to show that FCs of the first egg chamber derive from precursors that accumulate posterior to ICs and the germarium during the first half of pupation. Those studies also show that FSC precursors must reside within the germarium as ICs throughout pupation.

Figure 10: declining precursor proliferation from anterior to posterior during pupation, supporting lineage studies presented earlier in Table 2.

Figure 11: Repeat of targeted bond-GAL4 lineage studies with results compatible with the other lineage studies we presented.

We also started the Discussion with a summary of the most general insights.

Subsequent sections explain how the inherent difficulties of inferring cell origins in this paradigm were addressed, including inference of the locations of different precursors.

Aside from the general points described in the public review regarding the manuscript's presentation and the suggested genetic interrogation for model validation, we have the following suggestions:Figure panel labelling. We found the cartoons really helpful but we still got a bit lost with some of the figure panels and had to go back and forth between figures. It would be helpful to, for example, see the labels for specific markers on the figure panels themselves rather than having to check in each figure legend, and insert additional cartoons to orient the readers who might be unfamiliar with the system/cell types/markers.

We have added colored marker labels to most panels. We have made additional references to key cartoons, including adult germarium structures (Figure 1A), the summary cartoon of all lineage studies (Figure 2A), and the summary cartoon of inferred precursor progress through the first half of pupal development (Figure 5). The process of budding of the first egg chamber is directly evident from the fixed images presented in Figure 7 (even though the time-course is not of the same sample) and the live imaging frames in Figure 9 (and the video) and Figure 9- supplement 4 for the last phase, so we thought it unnecessary to add a cartoon of that process.

The authors only discussed the single cell RNAseq paper from the Lehmann lab in the context of bond-Gal4. Can anything else be learned by considering it more "holistically" in the context of the revised lineage relationships revealed by the authors? (e.g. regarding the separate precursors of FCs in different egg chambers, or the gradual restrictions of precursor fates according to cell locations along the AP axis…). Whilst the initial clustering may not have been revealing of these features, further (sub)clustering informed by this new knowledge may reveal some interesting candidate genes/cell subtypes.

The quoted studies report cell status in third instar larvae. We have shown that there are likely no definitive precursors for specific adult cell types at that stage. There will doubtless be AP differences in expression profiles amongst ICs that will correlate with more anterior or more posterior outcomes, most likely as a continuum. Those differences likely increase during pupation as precursor potentials become limited according to AP location; their identification would require RNA analysis at stages beyond pupariation. The Lehmann lab followed up their initial studies to target lineage analysis to Swarm cells as the earliest cells to migrate from apical locations, prior to pupation, as cited in the text.